# Senataxin resolves RNA:DNA hybrids forming at DNA double-strand breaks to prevent translocations

Sarah Cohen[1], Nadine Puget[1], Yea-Lih Lin[2], Thomas Clouaire[1], Marion Aguirrebengoa[1], Vincent Rocher[1], Philippe Pasero[2], Yvan Canitrot[1] & Gaëlle Legube[1]

Ataxia with oculomotor apraxia 2 (AOA-2) and amyotrophic lateral sclerosis (ALS4) are neurological disorders caused by mutations in the gene encoding for senataxin (SETX), a putative RNA:DNA helicase involved in transcription and in the maintenance of genome integrity. Here, using ChIP followed by high throughput sequencing (ChIP-seq), we report that senataxin is recruited at DNA double-strand breaks (DSBs) when they occur in transcriptionally active loci. Genome-wide mapping unveiled that RNA:DNA hybrids accumulate on DSB-flanking chromatin but display a narrow, DSB-induced, depletion near DNA ends coinciding with senataxin binding. Although neither required for resection nor for timely repair of DSBs, senataxin was found to promote Rad51 recruitment, to minimize illegitimate rejoining of distant DNA ends and to sustain cell viability following DSB production in active genes. Our data suggest that senataxin functions at DSBs in order to limit translocations and ensure cell viability, providing new insights on AOA2/ALS4 neuropathies.

[1] LBCMCP, Centre de Biologie Integrative (CBI), CNRS, Université de Toulouse, UT3, 118 Route de Narbonne, 31062 Toulouse, France. [2] Institut de Génétique Humaine, CNRS, Université de Montpellier, 34396 Montpellier, France. Correspondence and requests for materials should be addressed to G.L. (email: gaelle.legube@univ-tlse3.fr)

Mutations in the *SETX* gene are responsible for the rare neurological disorders ALS4, a dominantly inherited form of amyotrophic lateral sclerosis and ataxia with oculomotor apraxia type 2 (AOA2) which are associated with an early onset neurons degeneration (for review see ref. [1]). As a consequence, patients display strong ataxia as well as oculomotor troubles (AOA2) or muscle weakness (ALS4), generally occurring before their 30′s. *SETX* encodes a helicase, strongly conserved throughout evolution that has been implicated in a large variety of biological processes, from transcription termination, to meiosis completion and maintenance of genomic integrity[1]. At a molecular level, studies of the yeast senataxin homolog Sen1p established that this helicase displays an unwinding activity toward RNA:DNA hybrids[2–4]. Multiple genomic studies in both yeast and mammals recently unveiled that RNA:DNA hybrids mostly form as RNA polymerases progress throughout the genes, by the re-hybridization of the nascent RNA to the template DNA strand, leading to triple-stranded structure called R-loops (reviewed in ref. [5]). While R-loops display a strong ability to naturally form at GC-skewed promoters due to an enhanced thermodynamic stability of C-rich DNA: G-rich RNA duplexes[6,7], multiple complexes regulate their occurrence throughout the genome, including RNA splicing/processing factors and specific helicases, such as senataxin[8,9]. R-loops formation and processing play a crucial role in terminating transcription at least in yeast (reviewed in ref. [8]). In agreement, mutations in *sen1* trigger defective transcription termination and increased transcriptional read-through, especially on short transcribed units such as rDNA, tRNA, and small non-coding or coding genes (for review see ref. [10]). Such a function of R-loops processing and senataxin in transcriptional termination was also proposed to be conserved in higher eukaryotes, in a manner that would also involve dsRNA processing factors such as Drosha and DGCR8 as well as components of the RNA exosome (for review ref. [11]). However, beyond their role in regulating transcription, R-loops also represent a severe threat to genome integrity, proposed to arise both due to the susceptibility of the displaced single-stranded DNA to damaging agents as well as their potential to impede replication fork progression (reviewed in refs. [12–14]). In agreement, increased damage occurrence and genome instability was observed in cells deficient for R-loops processing factors such as AQR[15]. Additionally in yeast, *sen1* mutations are associated with an increased transcription-associated genome instability[16,17]. On the other hand, few studies also suggested that senataxin may play a more direct role at damage sites. Senataxin/Sen1p interacts with repair proteins in yeast and mammals and localizes to the site of damage during replication[18,19]. Moreover, *SETX* mutant mice display defective meiosis and Spo11-mediated DSB persistence[20]. Finally, depletion of senataxin/Sen1p triggers sensitivity to some DNA damaging agents such as $H_2O_2$ and UV[21–23], a feature also observed in AOA2 patient cell lines[22,24,25]. However, senataxin depleted cells are not radiation sensitive[22], suggesting that it may not function at sites of DNA double-strand break (DSB), a form of DNA damage largely induced upon irradiation.

Yet, recent studies have suggested that R-loops or/and RNA:DNA hybrids likely form at DSBs. An assay using a duplex specific nuclease detected RNA:DNA hybrids upstream the I-SceI site upon DSB formation[26]. Additionally a mutant form of RNAse H1, devoid of RNA exonuclease activity accumulates at the site of laser induced damages[27]. Finally, in *Schizosaccharomyces pombe*, RNA:DNA hybrids were shown to form during resection, regulating RPA filament formation[28]. The exact mechanism that leads to such R-loops or/and RNA:DNA hybrids formation remains unclear and may either relate to the transcriptional extinction observed at damaged sites or to de novo RNA PolII loading at DNA ends and subsequent RNA

production at the break point, two features that have been previously proposed to occur in many organisms (for review see ref. [12]).

In order to gain insights into R-loops biology at DSBs, here we set to assess a potential function of senataxin at sites of DNA DSBs. Using ChIP-seq and DRIP-seq, we uncovered that senataxin is recruited specifically at DSBs induced in transcriptionally active genes, which exhibit RNA:DNA hybrids accumulation following DSB induction. Senataxin distribution around DSBs coincided with a local decrease in R-loops and senataxin depletion triggered increased DSB-induced RNA:DNA hybrids formation, suggesting that senataxin processes DNA damage induced RNA:DNA hybrids. We found that senataxin is not required to sustain resection, nor rapid repair at these DSBs. Yet, it promotes Rad51 foci formation, counteracts translocations and sustains viability following DSB production in active genes, hence identifying a crucial and unanticipated role for senataxin in DSB repair.

## Results

**Senataxin is recruited at DSBs produced in active genes**. To assess senataxin recruitment at sites of DSBs, we used the DSB inducible via AsiSI (DIvA) cell line, which allows to induce clean DSBs throughout the genome[29,30]. In this cell line, 4 hydroxytamoxifen (4OHT) treatment induces the relocalisation of a stably expressed restriction enzyme (AsiSI) that in turn triggers the production of multiple DSBs at annotated positions across the genome, in a homogeneous manner in the cell population hence allowing the use of chromatin immunoprecipitation (ChIP)[30]. To obtain a quantitative and genome-wide assessment of senataxin binding and distribution at DSBs, we performed ChIP followed by high throughput sequencing (ChIP-seq) against senataxin before and after DSB induction in DIvA cells (respectively −4OHT and +4OHT). We observed an accumulation of senataxin in a 1–2 kb window surrounding AsiSI sites following 4OHT treatment (see examples in Fig. 1a). In vivo, AsiSI does not produce DSB at each of its 1211 annotated recognition sites, likely due to both DNA methylation and chromatin compaction[30]. Our recent studies allowed us to characterize AsiSI sites cleavage efficiency, using BLESS (direct in situ breaks labeling, enrichment on streptavidin and next generation sequencing) throughout the genome and to identify a set of 80 DSBs robustly induced following 4OHT treatment[31] (Clouaire, T. et al., manuscript submitted). Senataxin binding was significantly enriched following 4OHT on the AsiSI sites population that exhibits cleavage compared to the uncut AsiSI recognition sites (Fig. 1b). Heatmaps revealed that senataxin recruitment did not necessarily correlate with the cleavage efficiency, indicating that genomic and/or epigenomic features influence senataxin binding at DSBs (Fig. 1c).

Importantly, while AsiSI-induced DSBs mostly lie within promoters or gene bodies of active genes, some do reside in intergenic regions or in genes exhibiting no or very low level of RNA PolII (refs. [29,31] and see examples later). We previously reported that preexisting transcriptional activity strongly influences both DSB repair and signaling events[29–31]. Hence, we further tested whether DSB-induced senataxin recruitment may vary depending on the transcriptional activity of the broken locus. For this, we performed ChIP-seq mapping of the elongating form of RNA polymerase II (RNA PolII-S2P), of the total RNA PolII, as well as RNA-seq, prior DSB induction. DSBs were further sorted according to their transcriptional status prior to damage induction. Senataxin recruitment following DSB induction correlated with total RNA PolII enrichment levels preceding damage (Fig. 1d) as well as with elongating RNA PolII and RNA levels (Supplementary Fig. 1A, B). Moreover, inspection of

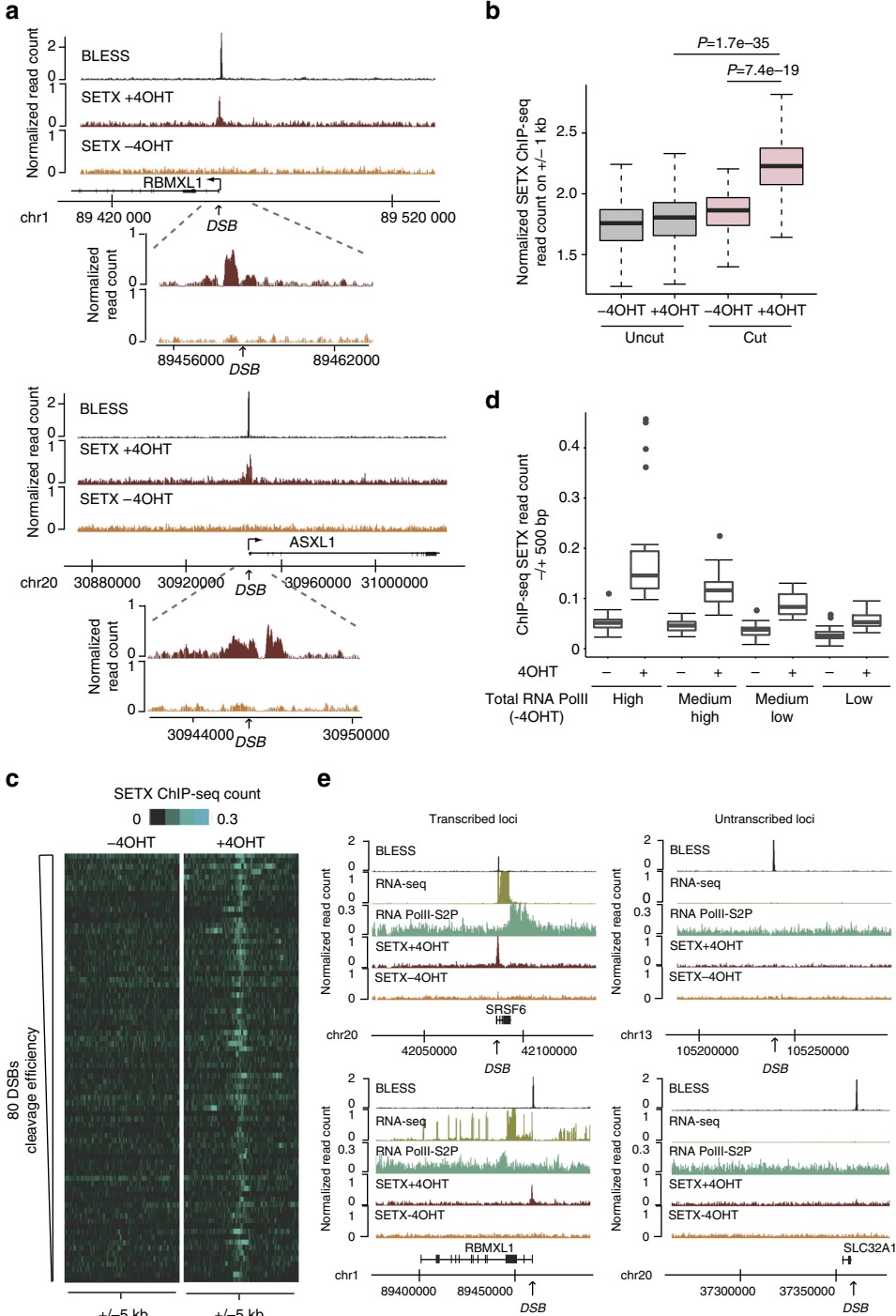

**Fig. 1** Senataxin is recruited at DSB induced in active loci. **a** Genome browser screenshots representing senataxin ChIP-Seq reads count before AsiSI activation (−4OHT) and after damage induction (+4OHT) at two individual AsiSI sites. The BLESS signal (indicative of cleavage efficiency (Clouaire, T. et al., manuscript submitted)) is also shown. Close up profiles are also shown below each screenshot. **b** Box plots representing senataxin ChIP-seq count before (−4OHT) and after (+4OHT) DSB induction at sites displaying AsiSI-induced cleavage ("cut", 80 AsiSI sites) or not ("uncut", 1139 AsiSI sites). Center line: median; box limits: 1st and 3rd quartiles; whiskers: maximum and minimum without outliers. P values are indicated (Wilcoxon Mann–Whitney test). **c** Heatmaps representing senataxin ChIP-seq count over a 10 kb window centered on the DSB before (−4OHT) and after (+4OHT) DSB induction. DSBs are sorted according to decreasing cleavage efficiency (based on BLESS data set (Clouaire, T. et al., manuscript submitted)). **d** Box plots representing senataxin ChIP-seq count before (−4OHT) and after (+4OHT) DSB induction at AsiSI "cut" sites sorted according to total RNA Polymerase II occupancy on a 10 kb window surrounding AsiSI sites (20 DSBs in each category). Center line: median; box limits: 1st and 3rd quartiles; whiskers: maximum and minimum without outliers. Points: outliers. **e** Genome browser screenshots representing senataxin ChIP-Seq reads count before AsiSI activation (−4OHT) and after damage induction (+4OHT) at four individual AsiSI sites, exhibiting either high (left) or low (right) transcriptional activity (indicated by RNA PolII S2P ChIP-seq mapping and RNA-seq −4OHT). The BLESS signal (Clouaire, T. et al., manuscript submitted) indicates that all sites display equivalent cleavage following 4OHT treatment

individual sites indicated that senataxin did not accumulate at broken intergenic or silent regions, although they were robustly cleaved (see BLESS signal) and showing high level of XRCC4 recruitment (assessed by ChIP-seq[29]) (Fig. 1e, Supplementary Fig. 1C). We previously demonstrated that transcriptionally active genes exhibit preferential binding of Rad51 and homologous recombination (HR) repair[29]. In agreement, senataxin displayed a significantly enhanced recruitment at DSB enriched in Rad51 compared to DSBs exhibiting low levels of Rad51 (Supplementary Fig. 1D). Altogether these data indicate that senataxin is recruited to damage sites with a strong preference for DSBs induced in transcriptionally active loci, preferentially repaired by HR.

**Senataxin removes DSB-induced RNA:DNA hybrids in active loci.** A well described function of senataxin is its ability to unwind RNA:DNA hybrids, hence regulating R-loops stability on the genome. Given senataxin recruitment at DSBs, we further investigated R-loops distribution at AsiSI-induced DSBs. For this, we performed DRIP-seq using the S9.6 antibody that displays a strong specificity for RNA:DNA hybrids[32,33] before and after DSB induction. As expected[6,7], DRIP-seq signal was enriched on active genes, peaking at TSS and TTS, validating our sequencing results (Supplementary Fig 2A, B). Notably, we could observe robust changes in R-loops distribution following damage, with an increase of RNA:DNA hybrids around the break site (Fig. 2a, Supplementary Fig. 2C). When taken collectively, following DSB induction, RNA:DNA hybrids were mildly but significantly enriched (7% increase) on a 10 kb window surrounding cut sites compared to uncut sites (Fig. 2b). Interestingly, while senataxin accumulation was strongly correlating with the transcriptional activity of the broken locus (Fig. 1 and Supplementary Fig. 1), this was less the case for RNA:DNA hybrids accumulation. Indeed, we could detect their formation upon damage at few (example Supplementary Fig. 2D, E)—but not all (example Supplementary Fig. 2F)—untranscribed loci (no detectable signal for RNA-seq or elongating RNA PolII). While total RNA PolII (hence likely not in an elongating form) was readily detectable prior DSB induction at some of these untranscribed loci (Supplementary Fig. 2D), others did not display any RNA PolII before damage induction (Supplementary Fig. 2E), suggesting that at least in a few instances RNA:DNA hybrids may also be able to form at untranscribed loci, devoid of RNA PolII (Discussion), albeit at lower levels.

Interestingly, at active genes, beyond the accumulation surrounding the DSBs, we could also detect a decrease in R-loops across damaged genes bodies and termination sites (Fig. 2a for examples, Supplementary Fig. 2G for averaged profiles). Moreover, careful examination of our high-resolution data revealed that despite an increase of RNA:DNA hybrids formation on ~10 kb window around DSBs, at active genes we could also observe a sharp 1–2 kb decrease of RNA:DNA hybrids at the exact sites of senataxin accumulation (Fig. 2c, d). Depletion of senataxin using a siRNA (Supplementary Fig. 3A) triggered an increase in DSB-dependent RNA:DNA hybrids accumulation proximally to a DSB (Fig. 2e). Thus, our high resolution data reveal that the pattern of R-loops shows complex alterations upon DSB induction. Altogether our data suggests that RNA:DNA hybrids form at DSB flanking chromatin and that, at active genes senataxin recruitment contribute to their removal in the immediate vicinity of DSBs.

**Senataxin promotes cell survival upon active genes breakage.** To further assess SETX function in DSB repair, we used our improved version of the DIvA cell line, whereby AsiSI-ER is also fused to an auxin inducible degron (AID). In this cell line, auxin addition triggers the degradation of the enzyme and hence repair of AsiSI-induced DSBs[29]. We first assessed the survival of AID DIvA cells following break induction and repair, in both control and senataxin-depleted cells (Supplementary Fig. 3A). Clonogenic assays revealed that depletion of senataxin using siRNA does not trigger cell death in absence of exogenous damage (Fig. 3a, −4OHT). DSB induction for 4 h, followed by auxin addition (+4OHT + auxin) only led to a mild survival defect in control cells (Fig. 3a) indicating that these cells recover well from the induction of DSBs by AsiSI, as previously reported[34]. In contrast, senataxin depleted cells exhibited a strong sensitivity to AsiSI-induced DSBs (Fig. 3a, Supplementary Fig. 3B).

Since AsiSI-induced DSBs exhibit clean DNA ends and undergo repeated cycles of cleavage, we further tested the effect of senataxin depletion at DSBs produced by other means. Etoposide is an inhibitor of Topoisomerase II (TOP II) and multiples studies have revealed that TOP II exerts a critical function at active genes in order to unwind supercoils and release topological constraints that form at transcriptionally active regions[35]. Hence, etoposide-induced DSBs exhibit a biased distribution throughout the genome, being preferentially located in active promoters (~30%) and genes bodies (~40%)[36]. Interestingly, senataxin depletion also triggered enhanced sensitivity to etoposide (Supplementary Fig. 3C). On the other hand, ionizing radiation (IR) induces DSBs randomly across the genome, hence being mainly located in intergenic loci that represent over 95% of higher eukaryotes genomes. Notably, and in agreement with a previous report[22], senataxin depletion did not trigger enhanced sensitivity to irradiation (Supplementary Fig. 3D). Our data therefore suggest that senataxin exerts an important function in cell survival specifically following break induction in active loci.

**Senataxin regulates γH2AX accumulation.** To further decipher the function of senataxin in DSB repair and to understand the lethality observed following DSB induction in senataxin deficient cells, we analyzed the effect of senataxin depletion on γH2AX foci formation following DSB induction. We found that senataxin depletion did not abolish foci formation and rather triggered enhanced γH2AX signal in 4OHT-treated cells (Fig. 3b). Interestingly, senataxin depletion also increased γH2AX signaling following etoposide treatment (Supplementary Fig. 3E) but not following IR (Supplementary Fig. 3F), suggesting a function of senataxin in regulating γH2AX foci formation at DSBs induced in active genes. To further strengthen these data, we tested whether a short global transcription extinction preceding break induction was able to rescue the increased γH2AX signaling observed in senataxin-depleted cells. A pretreatment of DIvA cells with cordycepin, a well characterized transcription inhibitor, partially reduced γH2AX in damaged DIvA cells following senataxin depletion (Fig. 3c). Altogether, our data suggest that senataxin is involved in regulating γH2AX establishment at DSBs induced in transcriptionally active loci.

We next investigated the consequences of senataxin depletion on DSB repair kinetics. Immunofluorescence performed at different time points after auxin addition revealed that γH2AX foci disappeared with the same kinetics in both control and SETX siRNA-treated cells (Fig. 4a, b). To refine this analysis, experiments were also performed using a high content microscope which allows to sort G1 versus G2 cells based on their DNA content[31]. Similarly we could not detect any delay of γH2AX foci clearance following auxin addition in SETX-depleted G1 and G2 cells (Supplementary Fig. 4A). Additionally, AID DIvA cells allow to assay repair kinetics at specific DSBs using a protocol based on

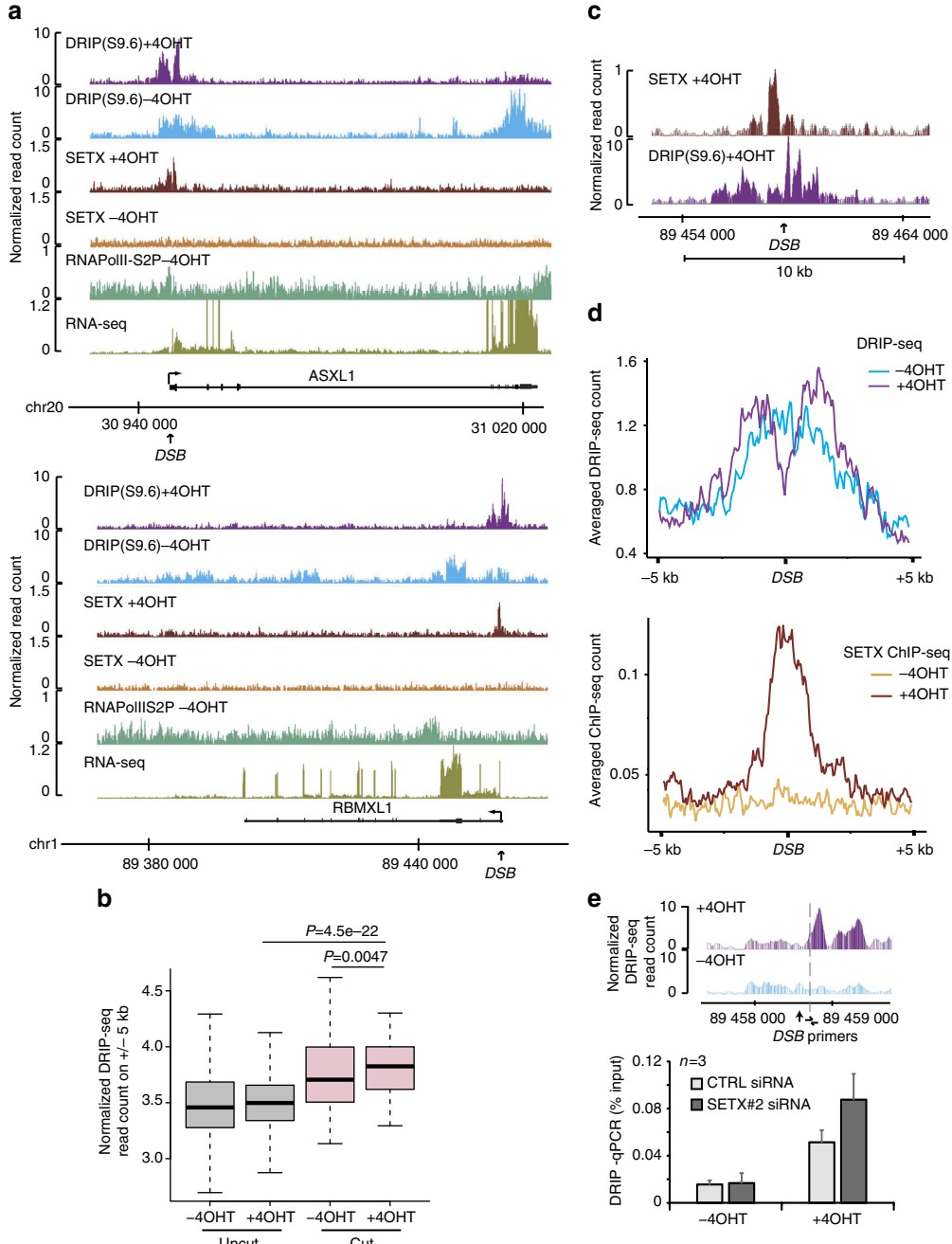

**Fig. 2** Senataxin removes DSB-induced RNA:DNA hybrids in active loci. **a** Genome browser screenshots representing DRIP-seq reads count before (−4OHT) and after damage induction (+4OHT) at two individual AsiSI sites. Senataxin profiles in both conditions are also shown, together with BLESS enrichment following DSB induction as well as RNA PolII-S2P and RNA-seq prior to DSB induction. **b** Box plots representing DRIP-seq reads count before (−4OHT) and after (+4OHT) DSB induction at sites displaying AsiSI-induced cleavage ("cut") or not ("uncut"). Center line: median; box limits: 1st and 3rd quartiles; whiskers: maximum and minimum without outliers. *P* values are indicated (Wilcoxon–Mann–Whitney test). **c** Close-up genome browser screenshot of senataxin and RNA:DNA hybrids at an individual DSB upon DSB induction. **d** Average DRIP-seq (top) and senataxin ChIP-seq (bottom) profiles on a ±5 kb window centered on the 80 AsiSI induced DSBs. **e** DRIP-qPCR in control (CTRL) and senataxin (SETX#2) depleted cells before (−4OHT) and after (+4OHT) DSB induction in DIvA cells as indicated. The position of the primers used to quantify RNA:DNA hybrids by qPCR are indicated on the genome browser screenshot above. Mean and s.e.m. of three biological replicates is shown

the ligation of a biotinylated oligonucleotide followed by streptavidin purification and quantitative PCR measurement of purified DNA[30,37]. Once more, senataxin depletion did not trigger repair delay at two DSBs found to be enriched in senataxin following 4OHT treatment (Fig. 4c). In addition, this held also true in G1- and G2-arrested cells following treatment with lovastatin and RO-3306 respectively (Supplementary Fig. 4B).

Altogether, these data indicate that while SETX deficiency triggers enhanced γH2AX signaling, it is not associated with delayed DSB repair, neither in G1 nor in G2.

**Senataxin regulates repair pathway choice.** Given that senataxin was found to be recruited at DSBs induced in transcriptionally

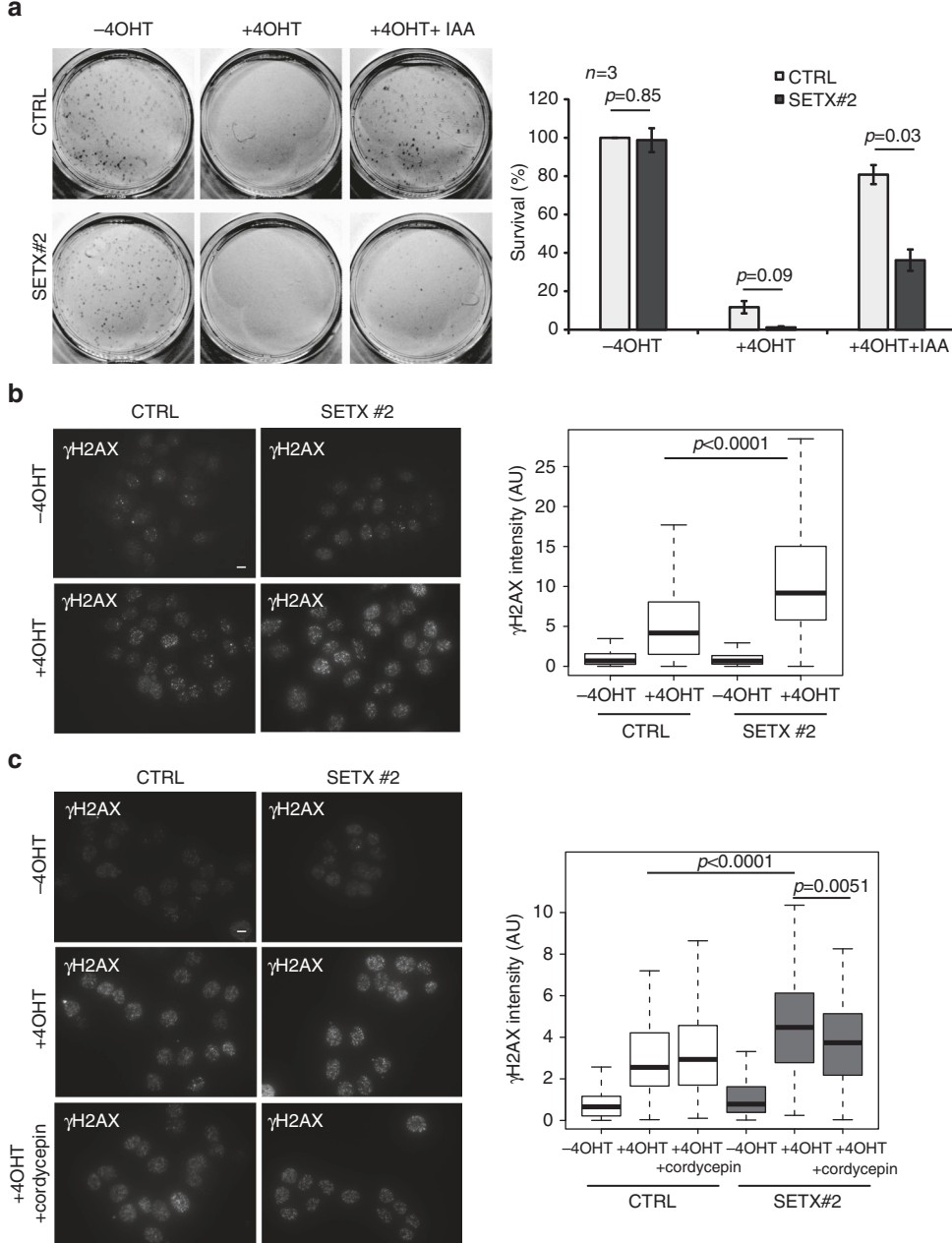

**Fig. 3** Senataxin regulates survival and γH2AX upon DSB induction. **a** Clonogenic assays in AID DIvA cells transfected with control and SETX siRNA, before and after 4OHT treatment (4 h), followed by auxin (IAA) treatment (4 h) as indicated. Left panel shows a representative experiment. Right panel shows the average and s.e.m. of three biological replicates. *P* values are indicated (paired *t*-test). **b** γH2AX staining performed in untreated or 4OHT-treated DIvA cells (4 h), after transfection with control or SETX siRNA as indicated. Scale bar: 10 μM. Right panel shows the quantification of the γH2AX nuclear signal within foci (>100 nuclei) from a representative experiment. Center line: median; box limits: 1st and 3rd quartiles; whiskers: maximum and minimum without outliers. *P* values are indicated (unpaired *t*-test). **c** γH2AX staining performed in control or SETX-siRNA transfected DIvA cells as indicated, treated with 4OHT or pretreated with cordycepin (1 h) previous 4OHT addition (4 h). Scale bar: 10 μM. Quantification is shown on the right panel (>100 nuclei, from a representative experiment). Center line: median; box limits: 1st and 3rd quartiles; whiskers: maximum and minimum without outliers. *P* values are indicated (unpaired *t*-test)

active loci, which are prone to undergo HR repair[29], we further investigated the function of senataxin on HR. Senataxin depletion impaired Rad51 foci formation, while increasing 53BP1 accumulation following DSB induction by AsiSI (Fig. 5a, b). Notably, the enhanced accumulation of 53BP1 observed in senataxin depleted cells was partially reversed by pretreating the cells with transcription inhibitors (cordycepin and 5,6-dichloro-1-β-D-ribofuranosyl-benzimidazole (DRB)) (Supplementary Fig. 5A, B). Next, to further investigate the function of senataxin in repair

pathway choice, we used the reporter constructs previously developed to quantitatively measure HR[38], single strand annealing (SSA)[39], and NHEJ[40], using flow cytometry following I-SceI transfection. Senataxin depletion triggered a mild decrease of HR and SSA associated with a similarly mild increase of NHEJ (Fig. 5c). Importantly, western blot against I-SceI (myc tagged) indicated that this was not due to changes in I-SceI expression.

Because generation of ssDNA is the initial step for HR repair, we further examined the effect of senataxin depletion on DSB end

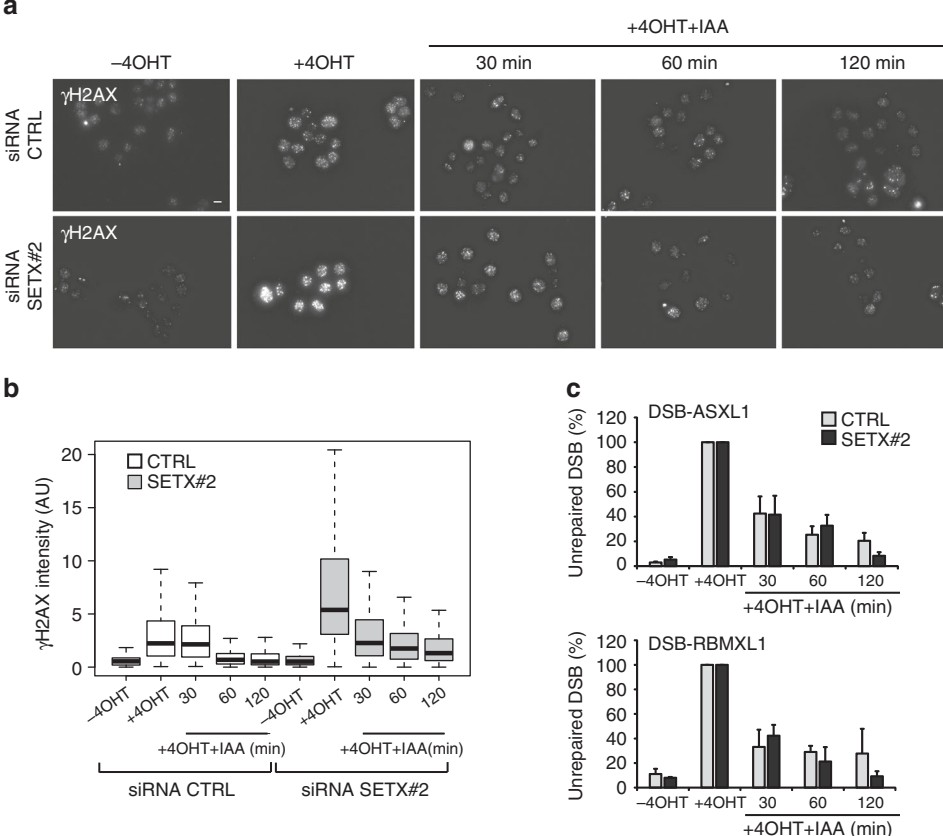

**Fig. 4** Senataxin depletion does not delay repair kinetics. **a** γH2AX staining performed in untreated or 4OHT-treated AID DIvA cells (4 h), followed by auxin (IAA) addition, after transfection with control or SETX siRNA as indicated. Scale bar: 10 μM. **b** Quantification of the γH2AX nuclear signal ( > 100 nuclei) from a representative experiment, performed in the above condition. Center line: median; box limits: 1st and 3rd quartiles; whiskers: maximum and minimum without outliers. P values are indicated (unpaired t-test). **c** Cleavage assay performed in AID-DIvA cells left untreated or treated with 4OHT (4 h) followed by auxin (IAA) addition (30 min, 60 min, and 120 min), after transfection of control or SETX-directed siRNA. Precipitated DNA was analyzed close to two DSBs, found to recruit SETX after 4OHT. The percentage of sites that remain broken for each DSBs after the indicated time of auxin treatment are presented. Average and s.e.m. (n = 3, biological replicates) are shown

resection. For this, we used an assay developed previously that allows to quantitatively measure single stranded DNA (ssDNA) generated at site specific DSBs[41] (Fig. 5d). Senataxin depletion did not reduce ssDNA levels at two DSBs induced by AsiSI (Fig. 5e) indicating that it is not necessary to promote resection.

Collectively these data indicate that senataxin promotes HR repair downstream of resection, by promoting Rad51 recruitment and counteracting 53BP1 accumulation.

**SETX depletion enhances translocations**. Given the strong requirement of SETX for survival upon DSB induction in active genes (Fig. 3), despite no clear delay in repair kinetics (Fig. 4), we next set to assess whether SETX could influence the quality of the repair reaction, more specifically the frequency of illegitimate rejoining between distant DSBs, involved in the generation of translocations. Using high resolution Capture Hi-C, we recently demonstrated that DSBs can cluster when induced on active genes and identified the molecular identity of AsiSI-induced DSBs brought into spatial proximity within nuclear foci[31]. Hence, based on this knowledge we developed an assay to accurately measure the illegitimate rejoining of closely clustered DSBs. We could detect rejoining between DSBs induced on the same chromosome (between *MIS12* and *TRIM37* as well as in *LINC0072* and *LYRM2*) (Fig. 6a), but also between DSBs induced on different chromosomes (*MIS12::LYRM2* and *TRIM37::RBMXL1*) (Supplementary Fig. 6A). Importantly, SETX depletion led to a highly

reproducible increase of all four translocations events compared to control cells (Fig. 6b, Supplementary Fig. 6B, C). Notably, this increase of translocations observed in senataxin depleted cells was partially rescued by a pretreatment with DRB (Fig. 6c) or upon overexpression of RNAseH1, which degrades RNA:DNA hybrids (Supplementary Fig. 6C). These data indicate that senataxin plays a key role in counteracting illegitimate rejoining of DSBs induced in loci ongoing active transcription.

## Discussion
In this study we set to better understand the formation of RNA: DNA hybrids at DSBs as well as the potential function of RNA: DNA hybrids removal factors in DSB repair, focusing on senataxin, a well characterized R-loops helicase. We discovered that senataxin is specifically recruited at DSBs induced in active loci, where it removes RNA:DNA hybrids forming *in cis* to broken loci. Senataxin is further required to regulate γH2AX signaling, to promote Rad51 loading and to minimize abnormal rejoining of distant DNA ends (Fig. 6d).

Our genome-wide mapping indicates that RNA:DNA hybrids accumulate *in cis* to DSBs, as previously proposed[26–28]. However, it seems that proximal DSB-induced RNA:DNA hybrids may form differently depending on the transcriptional status of the damaged locus. At active genes, this RNA:DNA hybrids accumulation around DSBs is associated with an otherwise R-loops decrease across the entire damaged gene body. Several studies

have reported that transcription is downregulated at the damaged gene, as well as on chromatin proximally flanking DSBs, although being globally maintained in the γH2AX domain at distance from the break (reviewed in refs. [12,42]). The exact mechanism leading to transcriptional repression *in cis* to DSBs is not yet clear but involves the recruitment of chromatin modifying complexes[43–47], as well as ATM-induced modifications of transcription elongation factors[45,47] yielding to a reduction in the elongating form of RNA PolII across the gene body[48]. Multiple studies have now established that R-loops accumulate at sites of paused or slowly

elongating RNA PolII[7,49]. Hence, at active genes, DSB-induced RNA PolII stalling may contribute to the strong RNA:DNA hybrids and/or R-loops formation in *cis* to the break.

Notably, although it was not a general feature (Supplementary Fig. 2F), we could also identify few seemingly transcriptionally silent sites that displayed low, but detectable levels of RNA:DNA hybrids following breakage (Supplementary Fig. 2D, E), even when total RNA PolII was not present prior damage (Supplementary Fig. 2E). Hence de novo RNA PolII recruitment at DNA ends (a feature recently observed[28,50]) may, at least in some

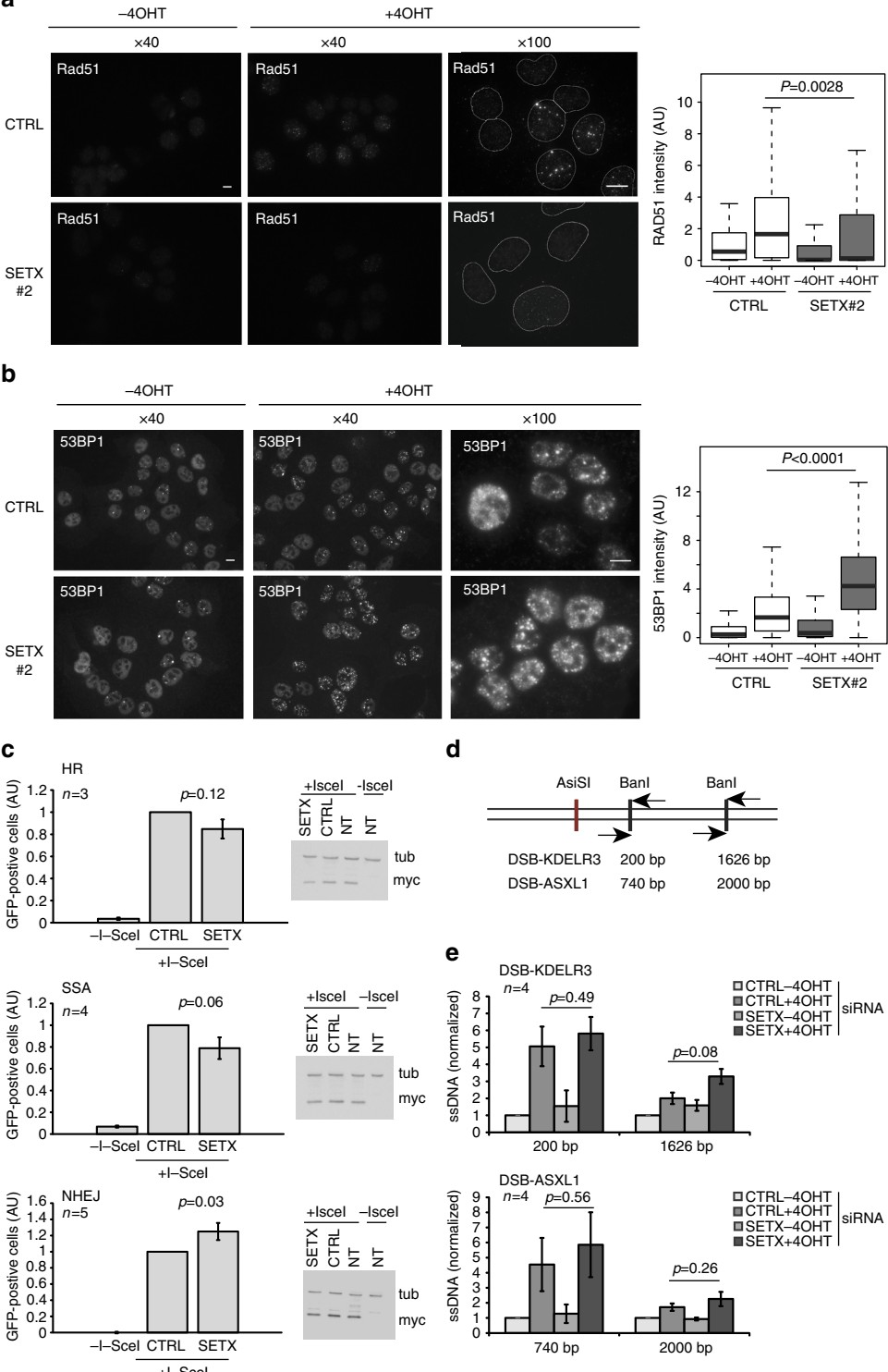

instance, produce RNA:DNA hybrids *in cis* to DSBs. More dedicated systems that allow to induce DSBs at a larger number of transcriptionally silent loci (using multiple guides RNAs for Cas9 induced breaks for example) should help to understand the occurrence of such RNA:DNA hybrids as well as the mechanisms driving their formation. Additionally, it is important to note here that our study does not allow to determine whether the S9.6 signal detected at DSBs represent triple stranded structures (R-loops) or double-stranded RNA:DNA hybrids. Indeed, in *S. pombe*, such hybrids were proposed to form as resection progresses, by the hybridization of a RNA to the resected single strand DNA[28]. Strand-specific mapping of the RNA engaged in the hybrids detected at DSB, by DRIPc-seq, should help to determine whether these resection-dependent RNA:DNA hybrids are conserved in mammalian cells and whether the increased S9.6 signal at DSBs observed in this study represent R-loops or RNA:DNA hybrids.

Regardless of whether they arise from an unscheduled activity of RNA Polymerase II blocked by the lesion, or by de novo transcription from DNA-ends, DSB-induced RNA:DNA hybrids may exert some important function in the DDR. First, these RNA:DNA hybrids could contribute to setup an adequate chromatin landscape. Indeed, R-loops at genes have been proposed to modify chromatin structure. For instance, R-loops trigger H3-S10 phosphorylation linked to chromatin condensation[51,52,]. On another hand, R-loops also coincide with increased chromatin accessibility[7] and can mediate the recruitment of the TIP60/p400 complex that promotes histone acetylation and nucleosome remodeling[53]. Notably, the Tip60 complex has been repeatedly found at break site, where it mediates H4 and H2A acetylation and subsequent recruitment of repair proteins[54–58]. Hence one can hypothesize that R-loops and/or RNA:DNA hybrids will contribute to set up the proper chromatin landscape required at DSBs to ensure accurate and timely repair. Second, RNA:DNA hybrids may contribute in promoting premature transcription termination of broken genes. Indeed, a large amount of studies established a function for R-loops, as well as for senataxin, in terminating transcription including for promoter-associated bidirectional non-coding RNA (for instance[59,60], reviewed in refs. [1,11]). In this regard it is interesting that dsRNA processing factors such as Drosha and Dicer contribute to this process[61–63]. Hence, at damaged genes, R-loops and senataxin may promote premature termination in order to allow either RNA PolII clearance from the damaged region to favor accessibility of repair proteins, or efficient recycling of RNA PolII to resume transcription after repair. It is tempting to speculate that the reported function of Dicer and Drosha in DDR[64] may at least in part, relate to the transcriptional termination at genes experiencing a DSB.

Interestingly we found that, although senataxin depletion did not delayed DSB repair, it impaired Rad51 foci formation, and conversely increased accrual of 53BP1. Decreased HR and increased NHEJ were also observed following senataxin depletion using HR and NHEJ reporter constructs although to a milder extent, which may be due to low R-loops formation on these substrates. Notably, senataxin depletion did not impede resection, suggesting that senataxin and/or RNA:DNA hybrids removal are critical at a step subsequent to ssDNA generation. However, resection was only monitored up to 1.6 kb from the DSB, so we cannot exclude a function of senataxin in regulating more long range resection events. Notably, we also found that senataxin regulates γH2AX establishment and counteracts the illegitimate rejoining of distant DNA ends, suggesting that R-loops removal is required to minimize translocations and maintain genome integrity following production of DSB in active genes. The mechanism by which senataxin impacts on γH2AX is currently unknown but may involve the regulation of ATM recruitment or activity. Alternatively, senataxin and/or R-loops may regulate the DSB-flanking chromatin structure, modifying its ability to undergo H2AX phosphorylation. Equally, how senataxin counteracts translocations needs to be investigated. We recently proposed that γH2AX spreading on entire topologically associated domains (TAD) likely modifies the properties of the chromatin fiber which could translate into changes in chromatin mobility within the nucleus[42,65]. Given that DSBs induced in active genes were found to display enhanced clustering ability[31], we can speculate that the increased aberrant joining of distant DSBs observed in senataxin depleted cells arise from an increased DNA ends mobility triggered by the enhanced γH2AX establishment.

Importantly, SETX is a gene mutated in two severe neurological diseases, AOA2 and ALS4, associated with progressive neurodegeneration. In this regard it is interesting that DSBs have been shown to be produced as a consequence of neuronal activity[66] and further genomic studies suggested they likely arise in active genes[67,68]. Given that senataxin exerts its anti-translocation and survival-promoting functions only for DSBs induced in active genes, we propose that this "Transcription Coupled DSB repair" function of senataxin may contribute to neuron loss in AOA2/ALS4 patients.

## Methods

**Cell culture.** U20S were retrieved from ATCC and modified with a plasmid encoding for the restriction enzyme (pBABE-AsiSIER and pAID-AsiSIER)[29,30]. U2OS, DIvA (AsiSI-ER-U20S), and AID-DIvA (AID-AsiSI-ER-U20S) cells were cultured in Dulbecco's modified Eagle's medium (DMEM) supplemented with antibiotics, 10% FCS (InVitrogen) and either 1 μg/mL puromycin (DIvA cells) or 800 μg/mL G418 (AID-DIvA cells) at 37 °C under a humidified atmosphere with 5% $CO_2$. The cell lines were regularly checked for mycoplasma contamination. For AsiSI-dependent DSB induction, cells were treated with 300 nM 4OHT (Sigma, H7904) for 4 h. When indicated, 4OHT-treated cells were washed three times in

**Fig. 5** Senataxin depletion decreases HR but does not impede resection. **a** Rad51 staining performed in untreated or 4OHT-treated DIvA cells (4 h), after transfection with control or SETX siRNA as indicated. Scale bar: 10 μM. Right panel shows the quantification of the Rad51 nuclear signal within foci (>100 nuclei) from a representative experiment. Center line: median; box limits: 1st and 3rd quartiles; whiskers: maximum and minimum without outliers. *P* values are indicated (unpaired *t*-test). **b** 53BP1 staining performed in untreated or 4OHT-treated DIvA cells (4 h), after transfection with control or SETX siRNA as indicated. Scale bar: 10 μM. Right panel shows the quantification of the 53BP1 nuclear signal within foci ( > 100 nuclei) from a representative experiment. Center line: median; box limits: 1st and 3rd quartiles; whiskers: maximum and minimum without outliers. *P* values are indicated (unpaired *t*-test). **c** HR (top panel), SSA (middle panel), and NHEJ (bottom panel) usage was measured using cell lines harboring specific reporter constructs[38–40] in control or senataxin-deficient (siRNA-transfected) dedicated cells. Myc-I-SceI expression was controlled by western blot in each condition. Mean and s.e.m. of at least three biological replicates are shown (as indicated). *P* values are indicated (paired *t*-test). **d** The site specific resection assay has been described earlier. Briefly, DNA purified from damaged or undamaged cells is digested by dedicated restriction enzymes (as indicated) and digestion-resistant DNA (single stranded DNA) is measured by qPCR, using primers pairs apart from the restriction sites. Here, we optimized this assay at two AsiSI-induced DSBs that were shown to undergo HR[29]. **e** Resection assay at the two DSBs in control or SETX-siRNA transfected cells. Values were normalized against the % of ssDNA detected in control cells before 4OHT treatment. Average and s.e.m. of four biological replicates are shown. *P* values are indicated (paired *t*-test)

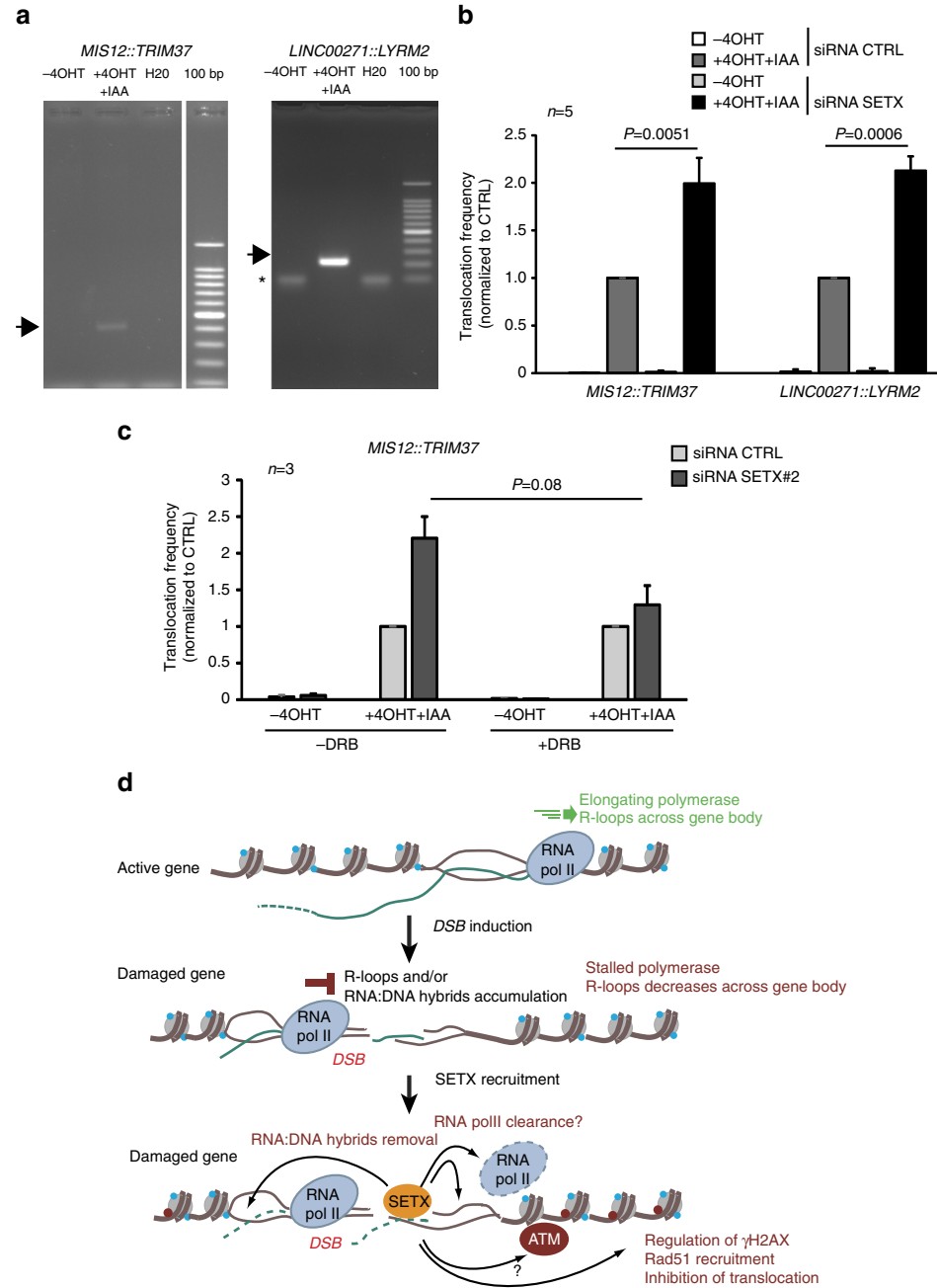

**Fig. 6** Senataxin counteracts the formation of translocations. **a** Rejoining of distant DSBs were detected by PCR, following DSB induction and repair (+4OHT + IAA 2 h) at breaks recently shown to undergo clustering[31]. DNA sequencing confirmed the nature of the amplified products. **b** *MIS12::TRIM37* and *LINC00271::LYRM2* rejoining frequencies were analyzed before or after 4OHT + IAA treatment, by quantitative PCR in AID DIvA cells transfected with control or SETX directed siRNA. Mean and s.e.m. of five biological replicates are shown. *P* values are indicated (one sample *t*-test). **c** *MIS12::TRIM37* rejoining frequency was analyzed in control or SETX-depleted AID DIvA cells pretreated or not with DRB prior to 4OHT addition as indicated. Mean and s. e.m. of three biological replicates are shown. *P* value is indicated (paired *t*-test). **d** Model: R-loops form as the RNA Polymerase II progresses across the gene. The induction of a DSB elicits ATM activity which triggers RNA polymerase II stalling at the vicinity of the DSB, hence decreasing R-loops across the gene body. On another hand, R-loops and/or RNA:DNA hybrids accumulate *in cis* to the DSB, due to stalled RNA PolII generating short, abortive, RNAs which thread back in the DNA duplex, or/and potentially de novo PolII transcription from DNA end. Senataxin is further recruited to remove RNA:DNA hybrids at the vicinity of the break induced in active loci. Senataxin and/or R-loop removal, regulate γH2AX establishment, promote Rad51 loading and minimize the occurrence of translocation by a mechanism that still need to be investigated

pre-warmed PBS and further incubated with 500 μg/mL auxin (IAA) (Sigma; I5148) for the indicated time. For transcriptional inhibition, DRB (Sigma, 100 μM) or cordycepin (Sigma, 50 μM) was added to the medium 1 h prior to 4OHT (4 h) and auxin (2 h) treatments. Cells were arrested in G1 using a 48 h treatment with 40 μM lovastatin (Mevinolin from LKT Laboratories) and in G2 with a 24 h treatment with 40 μM Ro-3306 (CDK1 inhibitor, Calbiochem). For clonogenic

assays in U2OS cells, DSBs were induced either by increasing doses of etoposide (Sigma) for 16 h as indicated or by irradiation with a Cs137 source (Biobeam 8000).

**siRNA and plasmid transfection**. siRNA transfections were performed with a Cell Line Nucleofector kit V (Program X-001, Amaxa) according to the manufacturer's instructions and cells were assayed 48 h after siRNA nucleofection. The following

siRNA against SETX were used: SETX#2 GAGAGAAUUAUUGCGUACU and SETX Smart pool (Dharmacon) containing the following siRNA: #a GCACGU-CAGUCAUGCGUAA, #b UAGCACAGGUUGUUAAUCA, #c AAAGAGUA-CUUCACGAAUU #d GGACAAAGAGUUCGAUAGA. siRNAs efficiency was assessed by mRNAs extraction with a Qiagen RNeasy kit (Qiagen) and reverse transcription with the AMV reverse transcriptase (Promega). cDNAs were quantified by RT-qPCR (primer sequences: SETX_FW CTTCATCCTCGGA-CATTTGAG and SETX_REV TTAATAATGGCACCACGCTTC) and normalized to RPLP0 cDNA levels (primer sequences: FW GGCGACCTGGAAGTCCAACT and REV CCATCAGCACCACAGCCTTC). For RNAse H1 overexpression, pICE-NLSmCherry and pICE-RNaseHI-NLS-mCherry were transfected 24 h after siRNA transfection using Lipofectamin 2000 (Life Technologies) following manufacturer's instructions.

**Western blot**. To assess for SETX depletion, western blot analysis was performed with NuPAGE Bis–Tris 4–12% gels and reagents (Invitrogen) according to the manufacturer's instructions. Briefly, cells were lysed in NuPage sample buffer with reducing agent (Invitrogen) and resolved proteins were transfered onto PVDF membranes (Invitrogen). PVDF membranes were then saturated 1 h in 5% nonfat dry milk with TBS and 0.5% Tween 20 and incubated overnight with the following primary antibodies: anti-SETX (Novus Biologicals, NB100-57542, 1:500) and anti-alpha-tubulin (Sigma-Aldrich, DM1A, 1: 100,000). Horseradish peroxidase-coupled secondary antibodies were from Sigma (anti-mouse, A2554, 1: 10,000; anti-rabbit, A0545, 1: 10,000), and the chemiluminescence Lumilight reagent was from Roche Diagnostic. To analyze I-SceI expression, total cell lysates were prepared 24 h after I-SceI plasmid transfection by direct resuspension of cells in Laemmli buffer and sonication. Cell extracts were separated on 10% SDS PAGE and proteins were transferred on nitrocellulose membrane. Primary antibodies were anti-myc (9E10, Roche) and anti-alpha-tubulin (Sigma-Aldrich). Signals were analyzed by autoradiography (for SETX expression) or using a ChemiDoc touch device (BioRad) for I-SceI expression.

**ChIP followed by high throughput sequencing and RNA-seq**. Cells were crosslinked with formaldehyde (1%) added to the culture medium for 15 min at room temperature. Glycin (0.125 M) was added for 5 min to stop the reaction. Cells were washed twice with cold PBS and harvested by scraping. Pelleted cells were incubated in lysis buffer (Pipes 50 mM pH 8, KCl 85 mM, NP-40 0.5%), homogenized with a Dounce homogenizer. Nuclei were harvested by centrifugation and incubated in nuclear lysis buffer: (50 mM Tris pH 8.1, 10 mM EDTA, 1% SDS). Samples were sonicated ten times for 10 s at a power setting of 5 and 50% duty cycle (Branson Sonifier 250), to obtain DNA fragments of about 500–1000 bp. After sonication, samples were diluted ten times in dilution buffer (0.01% SDS, 1.1% Triton X-100, 1.2 mM EDTA, 16.7 mM Tris pH 8.167 mM NaCl) and pre-cleared for 2 h with 100 µl of protein-A and protein-G beads (Sigma), previously blocked with 500 µg of BSA 2 h at 4 °C. Precleared samples were incubated overnight at 4 °C on a wheel with specific antibodies. For SETX ChIP, 200 µg of chromatin was immunoprecipitated by using 2 µg of anti-SETX (Novus Biologicals, NB100-57542). For RNA Pol II ChIP, 25 µg of chromatin was immunoprecipitated with 2 µg of anti-RNA polymerase II CTD repeat YSPTSPS (phospho S2) (Chromotek 3E10), or with 2 µg of the anti-total RNA PolII (Bethyl Laboratories A304-405A). XRCC4 ChIP-seq were published earlier[29]. Immune complexes were precipitated with 100 µl of blocked protein A/protein G beads for 2 h at 4 C on a rotating wheel. Beads were washed with dialysis buffer (2 mM EDTA, 50 mM Tris pH 8.1, 0.2% Sarkosyl) once and with wash buffer (100 mM Tris pH 8.8, 500 mM LiCl, 1% NP-40, 1% NaDoc) four times. Immunoprecipitated complexes were resuspended in 200 µl of TE buffer (Tris 10 mM pH8, EDTA 0.5 mM pH8) with 30 µg of RNAse A for 30 min at 37 °C. Crosslink was reversed in the presence of 0.5% SDS at 70 °C overnight with shaking. After a 2 h proteinase K treatment, immunoprecipitated and input DNA were purified with phenol/chloroform and precipitated. Samples were resuspended in 100 µl water. For ChIP-Seq, multiple ChIP experiments were pooled. Immunoprecipitated DNA was subjected to library preparation and single-end sequencing on a NextSeq 500 at EMBL GeneCore (Heidelberg, Germany).

**RNA extraction and RNA-seq library preparation**. For RNA-seq, DIvA cells (transfected with the control siRNA) were lysed using TRI reagent (SIGMA) and spiked-in with ERCC RNA Spike-In Mix (Thermo Fisher Scientific). Total RNA were recovered by chloroform extraction followed by isopropanol precipitation. Samples were treated with RQ1 RNase-free DNase (Promega) for 1 h at 37 °C and purified by phenol/chloroform extraction followed by ethanol precipitation. Ribosomal RNA depletion and RNA-Seq library preparation were performed at EMBL Genomics core facilities (Heidelberg, Germany) using TruSeq Stranded Total RNA (Illumina).

**DNA:RNA immunoprecipitation**. DRIP assay was carried out according to the protocol described in ref. [7]. DIvA cells were treated with 300 nM 4OHT for 4 h, trypsinized, pelleted at low speed and washed with DPBS (Life technology). Total nucleic acids were extracted with 0.5% SDS /Proteinase K (Thermo Fisher Scientific, Waltham, MA) treatment at 37 °C overnight and recovered by phenol-

chloroform extraction and ethanol precipitation. DNA was digested by a restriction enzyme cocktail (20 units each of EcoRI, HindIII, BsrGI, XbaI) (New England Biolabs) in 1× NEBuffer 2, with or without RNase H treatment, overnight at 37 °C. Fragmented DNA was cleaned by phenol-chloroform extraction and ethanol precipitation followed by two washes with 70% ethanol. Air-dried pellets were resuspended in 10 mM Tris-HCl pH 7.5, 1 mM EDTA (TE). In total, 4 µg of digests was diluted in 450 µL of TE, and 10 µL was reserved as input for qPCR. 50 µL of 10× IP buffer was added (final buffer concentration of 10 mM sodium phosphate, 140 mM sodium chloride, 0.05% Triton X-100) and 10 µL of S9.6 antibody (1 mg/ml, kind gift from F. Chedin, UC Davis). Samples were incubated with the antibody at 4 °C for 2 h on a wheel. 50 µL of Protein A/G Agarose (Pierce), previously washed twice with 700 µL of 1× IP buffer for 5 min at room temperature, were added and samples were incubated for 2 h at 4 °C on a wheel. Each DRIP was then washed three times with 700 µL 1× IP buffer for 10 min at room temperature. After the final wash, beads were resuspended in 250 µL of 1× IP buffer and incubated with 60 units of Proteinase K for 45 min at 55 °C. Digested DRIP samples were then cleaned with phenol-chloroform extraction and ethanol precipitation. Air-dried DRIP pellets were resuspended in 45 µL of 10 mM Tris-HCl pH 8. DRIP experiment was assayed by qPCR using primers located at SNRNP (negative control), RPLA13 (positive control) and RBXML1 (AsiSI site):

SNRNP_FW:GCCAAATGAGTGAGGATGGT; SNRNP_REV: TCCTCTCTGCCTGACTCCAT;
RPLA13_FW:AATGTGGCATTTCCTTCTCG;
RPLA13_REV: CCAATTCGGCCAAGACTCTA.
RBMXL1_FW: GATTGGCTATGGGTGTGGAC
RBMXL1_REV: CATCCTTGCAAACCAGTCCT
For DRIP-seq, samples from three DRIP experiments were pooled and sonicated to an average size of 300 bp using a Bioruptor (Diagenode) for 20 cycles of 30 s on, 30 s off, high setting. Immunoprecipitated DNA was subjected to library preparation and single-end sequencing on a NextSeq 500 at EMBL GeneCore (Heidelberg, Germany).

**ChIP-seq, RNA-seq, and DRIP-seq data set analyses**. SETX ChIP-Seq, RNA-seq, and DRIP-Seq samples were sequenced using Illumina NextSeq 500 (single-end, 80-bp reads for SETX ChIP-Seq; paired-end, 75-bp reads for RNA-Seq and single-end, 85 bp reads for DRIP-Seq) at EMBL Genomics core facilities (Heidelberg, Germany). The quality of each raw sequencing file (fastq) was verified with FastQC (https://www.bioinformatics.babraham.ac.uk/projects/fastqc/). ChIP-Seq and DRIP-Seq files were aligned to the reference human genome (hg19) and processed using a classical ChIP-seq pipeline: bwa (http://bio-bwa.sourceforge.net/) for mapping and samtools (http://www.htslib.org/) for duplicate removal (rmdup), sorting (sort), and indexing (index). RNA-seq was mapped to a custom human genome (hg19 merge with ERCC92 sequences) to avoid mapping ERCC sequences on the human genome and processed as the same way as ChIP-Seq, except for the alignment in paired-end mode with STAR and without remove potential duplicate.

Coverage for each aligned ChIP-seq data set (.bam) were computed with the rtracklayer R package and normalized using total read count for each sample. Coverage data was exported as bigwig (file format) for further processing.

Averaged ChIP-seq, RNA-seq, and DRIP-Seq profiles were generated using the R package ggplot2. For profiles relative to DSB, the x-axis represents genomic position relative to AsiSI site and the y-axis represents the mean coverage at each bp (Fig. 2d). For metagene profiles, the mean coverage was computed in two parts: first, mean coverage was computed in 200 bp intervals 3 kb upstream TSS and downstream TSS. Second, gene bodies were divided into 100 equally sized bins, so average profiles could be computed as a percent of entire gene length. Profiles were computed for all genes (Supplementary Fig. 2B) or for genes either directly damaged or located near a DSB (<1 kb) (Supplementary Fig. 2G).

To classify DSBs based on transcriptional activity, RNA PolII-S2P ChIP-seq, total RNA PolII ChIP-seq or RNA-seq obtained in DIvA cells prior DSB induction were computed on a windows of ±5 kbp around DSBs. DSBs were ordered based on their RNA PolII enrichment and discriminated into four categories of 20 DSBs each (low, medium low, medium high, and high).

Box-plots were generated with R-base. The center line represents the median, box ends represent respectively the first and third quartiles, and whiskers represent the minimum and maximum values without outliers. Outliers were defined as first quartile −(1.5 × interquartile range) and above third quartile + (1.5 × interquartile range). Values represent the total normalized read count in a specific genomic window surrounding AsiSI-induced DSBs or uncut AsiSI genomic sites ("uncut"). Statistical hypothesis testing was performed using nonparametric paired Mann−Whitney−Wilcoxon (wilcoxon.test() function in R) to tests distribution differences between two populations.

For heatmap representations, average normalized sequencing signal was determined in 500 bp bins centered on each cleaved AsiSI site using custom R/Bioconductor scripts. The resulting matrix was represented as a heatmap using Java Treeview (http://www.jtreeview.sourceforge.net). DSBs were ordered based on the BLESS signal (Fig. 1c) or the total RNA PolII enrichment on ±5 kb (Supplementary Fig. 1B).

**Clonogenic assays**. After siRNA transfection, AID-DIvA and U20S cells were seeded at a clonal density in 10 cm diameter dishes. After 48 h, U20S cells were

either exposed at increasing doses of IR or treated with increasing doses of etoposide, as indicated. AID-DIvA cells were treated with 300 nM 4OHT for 4 h and, when indicated, washed three times in pre-warmed PBS and further incubated with 500 μg/mL auxin for another 4 h. After three washes in pre-warmed PBS, complete medium was added to each AID-DIvA cells dish. After 10 days, U20S cells and AID-DIvA cells were stained with crystal violet (Sigma) and counted. Only colonies containing more than 50 cells were scored.

**Immunofluorescence**. Cells were plated in glass coverslips, fixed with 4% paraformaldehyde for 15 min, permeabilized with Triton 0.5%, and blocked with PBS-BSA 3% for 30 min at room temperature. Cells were then incubated with antibody against γH2AX (JBW301, 05-636, Millipore, 1:1000) and 53BP1 (Novus Biological NB-300-104, 1:500) overnight at 4 °C. Cells were washed three times in PBS-BSA 3% and incubated with secondary antibody for 1 h. After three washes (one PBS-BSA 3% and two PBS), nuclei were stained with Hoechst 33342 (Sigma). Staining against Rad51 (Santa cruz sc8349, 1:200) was performed using the following protocol. Cells were plated in glass corverslips, then submitted to a pre-extraction with ice cold buffer (20 mM HEPES pH 7.5; 20 mM NaCl; 5 mM MgCl$_2$; 1 mM DTT; 0.5% NP40) for 20 min on ice. Cells were fixed with 4% paraformaldehyde for 15 min and blocked with PBS-BSA 3% for 30 min at room temperature. Immunofluorescence was carried as previously described. Image acquisition was performed using MetaMorph on a wide-field microscope equiped with a cooled charge-coupled device camera (CoolSNAP HQ2), using a ×40 or ×100 objective.

**High-throughput microscopy**. AID-DIvA and U20S cells were plated in 96-well plates after transfection. After fixation, permeabilization and saturation steps, γ-H2AX was stained overnight with γH2AX (JBW301, 05-636, Millipore) and the secondary antibody anti-mouse Alexa 647 (A21235, Molecular Probes). Nuclei were labeled with Hoechst 33342 (Sigma) at a final concentration of 1 μg/ml for 5 min. γ-H2AX foci were further analyzed with an Operetta automated high-content screening microscope (PerkinElmer). For quantitative image analysis, multiple fields per well were acquired with a ×40 objective lens to visualize ~2000 cells per well in triplicate.

**γ-H2AX, 53BP1, and Rad51 foci intensity quantification**. All quantification was performed using Columbus, the integrated software to the Operetta automated high-content screening microscope (PerkinElmer). DAPI or Hoechst nuclei were selected according to the B method, and appropriate parameters, such as the size and intensity of fluorescent objects, were applied to eliminate false-positive. Then γ-H2AX, Rad51 and 53BP1 foci were detected with the D method with adjusted parameters to ensure best foci detection: detection sensitivity 0.5–1; splitting coefficient, 0.5–1; background correction, >0.5–0.9. G1 and G2 nuclei were selected on the basis of the Hoechst intensity, after visualization of the Hoechst distribution in all cells. Box plots represent the total nuclear signal intensity detected in foci.

**Repair kinetics at AsiSI sites**. Repair kinetics at specific AsiSI-induced DSBs were measured as described in refs. [29,37]. Genomic DNA was extracted using the DNAeasy kit (Qiagen) and in vitro ligation with a biotinylated double-stranded oligonucleotide, ligatable with AsiSI sites, was carried out overnight at 4 °C. T4 ligase was inactivated at 65 °C for 10 min, then ligated DNA was fragmented by EcoRI digestion at 37 °C for 2 h. Digestion was then inactivated at 70 °C for 20 min. Samples were precleared with protein A beads for 2 h at 4 °C on a wheel. Precleared samples were then incubated with streptavidin beads (Sigma) at 4 °C overnight. Beads were previously saturated with 500 μg of BSA 2 h at 4 °C. DNA pulled down with streptavidin beads was washed once with dialysis buffer (2 mM EDTA, 50 mM Tris pH 8.1, 0.2% Sarkosyl), five times with wash buffer (100 mM Tris pH 8.8, 500 mM LiCl, 1% NP-40, 1% NaDoc), and three times in TE buffer (Tris10mM pH8, EDTA0.5 mM pH8). Beads were resuspended in 100 μL of water and digested with HindIII at 37 °C for 4 h. After phenol/chloroform purification and precipitation, DNA was resuspended in 100 μL water. qPCR was performed using the following primers: ASXL1-FW CCTAGCTGAGGTCGGTGCTA; ASXL1-REV GAA-GAGTGAGGAGGGGGAGT; RBMXL1-FW GATTGGCTATGGGTGTGGAC; RBMXL1-REV CATCCTTGCAAACCAGTCCT.

**Resection assay**. Measure of resection was performed as described in ref. [41] with the following modifications. DNA was extracted from fresh cells using the DNAeasy kit (Qiagen). In total, 400 ng were digested overnight at 37 °C using the Ban I restriction enzyme (16U per samples) that cuts at ~200 bp and 1626 bp from the DSB-KDELR3 and at 740 bp and 2000 bp for DSB-ASXL1. Digested and undigested samples were also treated with Rnase H (Promega). Ban1 was heat inactivated 20 min at 65 °C. Digested and undigested DNA were analyzed by qPCR using the following primers:
DSB-KDELR3_200 FW: ACCATGAACGTGTTCCGAAT;
DSB-KDELR3_200_REV: GAGCTCCGCAAAGTTTCAAG;
DSB-KDELR3_1626_FW: CCCTGGTGAGGGGAGAATC;
DSB-KDELR3_1626_REV: GCTGTCCGGGCTGTATTCTA;
DSB-ASXL1_740 FW: GTCCCCTCCCCCACTATTT;
DSB-ASXL1_740_REV: ACGCACCTGGTTTAGATTGG;
DSB-ASXL1_2000_FW: GTTCCTGTTATGCGGGTGTT;

DSB-ASXL1_2000_REV: TGGACCCCAAATTCCTAAAG.

ssDNA% was calculated with the following equation: $ssDNA\% = 1/(2^{(\text{Ct digested}-\text{Ct undigested}-1)}+0.5)*100$.

**HR, NHEJ, and SSA repair assays**. The GCS5 and RG37 cell lines have been derived from SV40 T-transformed human fibroblasts (GM639)[38,40]. The U2OS SSA has been derived from the osteosarcoma cell line U2OS[39]. For siRNA transfection, $1 \times 10^5$ cells were transfected using Interferin (Ozyme, France) with 10 nM of siRNA according to the manufacturer's instructions. Plasmid coding for I-SceI expression was transfected using JetPei (Ozyme, France) according to the manufacturer's instructions. Twenty four hours after siRNA transfection, cells were washed and transfected with the I-SceI coding plasmid (1 μg). After 72 h, cells were collected after trypsin treatment and analyzed by flow cytometry (BD Facscalibur) to detect and count GFP-positive cells in each condition. Percentage of GFP-positive cells was calculated on 25,000 sorted events.

**Translocation assay**. AID-DIvA cells were treated as indicated and then DNA was extracted from fresh cells using the DNAeasy kit (Qiagen). Illegitimate rejoining frequencies between MIS12 and TRIM37 (chr17_5390209 and chr17_57184285), LINC00217 and LYRM2 (chr6_135819337 and chr6_9034817), MIS12 and LYRM2 (chr17_5390209 and chr6_9034817), or TRIM37 and RBMXL1 (chr17_57184285 and chr1_89433139) were assessed by qPCR using the following primers:
MIS12_Fw: GACTGGCATAAGCGTCTTCG
TRIM37_Rev: TCTGAAGTCTGCGCTTTCCA
LINC00217_ Fw: GGAAGCCGCCCAGAATAAGA
LYRM2_Rev: TCTGAAGTCTGCGCTTTCCA
TRIM37_Fw: AATTCGCAAACACCAACCGT
RBMXL1_Rev: GCCAATGGAGTTCCCTGAGTC
Results were normalized using two control regions, both far from any AsiSI sites and γH2AX domain using the following primers:
Ctrl_chr1_82844750_Fw: AGCACATGGGATTTTGCAGG
Ctrl_chr1_82844992_Rev: TTCCCTCCTTTGTGTCACCA
Ctrl_chr17_9784962_Fw: ACAGTGGGAGACAGAAGAGC
Ctrl_chr17_9785135_Rev: CTCCATCATCGGCACCCTTTG.
Normalized translocation frequencies were calculated using the DeltaDeltaCt method from Bio-Rad CFX Manager 3.1 software[69].

**Data availability**. High throughput sequencing data have been deposited to Array Express under accession number E-MTAB-6318. Other data and source codes are available upon request.

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

## Acknowledgements

We thank the genomic core facility of EMBL for high throughput sequencing. We are grateful to the Non-Invasive Exploration service—US006 for the access to the irradiator (Biobeam 8000). We thank Dr. Lionel Sanz and Dr. Frederic Chedin (UC Davis) for providing advice for DRIP experiments. We thank Dr. Bernard Lopez (IGR Paris) for the NHEJ and HR reporter constructs, as well as Dr. Jeremy Stark (City of Hope) for the SSA reporter construct. pICE-NLSmCherry and pICE-RNaseHI-NLS-mCherry were a kind gift from Dr. S. Britton (IPBS Toulouse) We thank Dr. M. Bushell (University of Leicester) for helpful discussions and data exchange. We also thank Dr D'adda di Fagagna (IFOM) for sharing unpublished work. M.A. was supported by the Fondation pour la Recherche Médicale (FRM). Funding in GL laboratory was provided by grants from the European Research Council (ERC-2014-CoG 647344), Agence Nationale pour la Recherche (ANR-14-CE10-0002-01and ANR-13-BSV8-0013), the Institut National contre le Cancer (INCA), and the Ligue Nationale contre le Cancer (LNCC, "équipe labelisée").

## Author contributions

S.C., N.P., T.C., and Y.C. performed experiments. V.R., M.A., and T.C. performed bioinformatic analyses of ChIP and DRIP data sets. Y.-L.L. and P.P. provided technical and conceptual assistance for DRIP experiments. G.L. conceived, supervised, analyzed experiments and wrote the manuscript. All authors commented and edited the manuscript.

## Additional information

**Competing interests:** The authors declare no competing financial interests.

