## [Peer Review File · Nature Communications]

Reviewers' Comments:

Reviewer #1:

Remarks to the Author:

Summary:

Using their previously developed cell line in which up to 80 sequence-specific DSBs cells can be induced at specific sites, mainly active transcription units, in the genome, Cohen et al, assessed the role of R-loops in DSB response. Using this system, the authors showed that R-loops and the likely RNA:DNA helicase Senataxin (SETX) accumulate on chromatin flanking DSBs induced within transcribed genes. Interestingly, SETX depletion did not affect resection kinetics at DSBs, had minimal affect on DSB repair pathway choice and did not delay the kinetics of DSB repair. Surprisingly, and through a yet to characterised mechanism, they observed increased joining of distant DSBs to form translocations. Therefore, this study elucidates a new role for Senataxin and R-loops in regulating the fidelity of in DSB repair in mammalian cells. It also provides a possible resolution to the controversy about whether small dsRNAs are produced at DSBs. As small amounts of dsRNA has previously been reported to occur at transcription termination sites, which are also enriched in R-loops, the authors suggest that the so-called DDRNAs previously reported by others could simply result from pausing of RNA polIII at DSBs. In this respect pausing at DSBs really corresponds to premature transcription termination at DSBs within active transcription units.

Major points:

1. To assess the role of SETX in the choice or repair pathway (NHEJ versus HR), they authors use reporter systems based on GFP expression following DSB repair (Figure 5). However, it is not clear if R-loops are formed in these systems, as not all transcription units are known to produce R-loops. The authors should discuss whether formation of R-loops was previously shown in these specific reporter systems and, if not, either confirm R-loop formation experimentally or discuss this caveat to their experiments. If R-loops are not formed, then using these repair systems would not address the role of SETX in the choice of repair pathways. This could in fact understate the role of SETX in DSB repair pathway choice.
2. The authors have established an elegant assay to examine translocation events that occur between DSBs whose precise position on chromosomes is known (Figure 6). This provides sequence resolution of translocation events. It is an original approach presented for the first time in this paper. The data are strong but it would be useful to validate this method using an alternative technique, such as FISH. If the translocation events measured occur at a reasonable frequency, at least a few percent of the population, it might be possible to detect using FISH. Alternatively, if these events are rare it might be too much to expect their detection by this technology.
3. Should include a SETX cDNA rescue of the SETX knockdown as a control for one of the key experiments experiments of the paper, e.g. the translocation assay.
4. Overall the English needs to be improved by careful editing (see some suggestions below).

Minor points:

1. Legend to Figure 1. Regarding the BLESS dataset, it is not appropriate to refer to another manuscript in revision with a legend.
2. Spell out 'DSB Inducible via AsiSI' before first use of jargon acronym, DIVA.
3. Figure 1A. Provide zoomed-in view of both genome browser shots around DSBs. Histogramme for RBMXL1 gene looks minor in this view.
4. Figure 1E could usefully include a positive control, something recruited to both DSBs. However, if such a data set is unavailable then this is not a major flaw overall.
5. Page 7, first line. Important to state fold enrichment here. It appears to be statistically significantly but what is the fold enrichment measured. Appears to be quite small, 20% increase over uncut at best. Overall the data in Figure 2 is very nice.
6. Figure 3B. Show quantification of gH2AX signal from -4OHT cells.
7. Page 9, 4 lines from top. More accurate to state that DRB "... partially rescued the increased gH2AX foci observed following SETX depletion."
8. Page 10, 10 lines down. "... associated with a similarly mild increase of NHEJ...".

9. Regarding the required thorough editing of the language of the manuscript. One of the researchers in my lab that I consulted for advice on this manuscript went to the trouble of listing some suggested changes which I reproduce here for the benefit of the authors....

- P.1/title "broken genes repair" better: "repair of broken genes"

- Page 2:

o Line 8: SETX is the gene, so should this not be senataxin (as the protein) rather than the gene name?

o Line 8: recruited to DSBs, when they occur in transcriptionally active loci

o Line 13: found to be essential

o Line 14: Our data suggests that senataxin functions

- Page 3:

o Line 1-3: ALS4, a dominantly.....sclerosis, and ataxia with oculomotor apraxia type 2 (AOA2), which are associated with an early onset neurons degeneration

o Line 11: re-hybridisation

o Line 12: triple-stranded

o Line 20: R-Loops processing

o Line 24: threat for the genome

- Page 4:

o Paragraph 1/Line 4: R-Loops processing

o Paragraph 1/Line 5: with an increased

o Paragraph 1/Line 6: on the other hand

o Paragraph 2/Line 1: recent studies have recently suggested

o Paragraph 2/Line 5: what do you mean by "resection progresses"?

o Paragraph 2/Line 6: R-Loops accumulation

o Paragraph 3/Line 1: insights into R-Loops biology

o Paragraph 3/Line 3: which are preferred sites of preferential R-Loops accumulation

- Page 5:

o Paragraph 2/Line 2: 4-hydroxytamoxifen (4OHT) treatment (4OHT)

o Paragraph 2/Line 15: "cut AsiSI sites population" rephrase

- Page 6:

o Paragraph 2/Line 7-9: Is RNAPolIII required for SETX recruitment or vice versa?

o Paragraph 2/Line 11: Altogether, this data indicates that

o Paragraph 2/Line 12: recruited to damage sites

o Paragraph 3/Line 2: R-Loops stability

o Paragraph 3/Line 3: R-Loops distribution

o Paragraph 3/Line 7: R-Loops distribution

- Page 7:

o Paragraph 1/Line 2: R-Loops formation

o Paragraph 1/Line 6/7: Altogether, this data shows that

o Paragraph 2/Line 2: R-Loops formation

o Paragraph 2/Line 4: R-Loops removal

o Paragraph 3/Line 3 triggers the enzyme degradation

- Page 8:

o Paragraph 2/Line 3: that TOP11 exerts

o Paragraph 2/Line 8: On the other hand, Irradiation (IR) induces

o Figure S3A: tubulin is overexposed

- Page 9:

- o Paragraph 1/Line 5: our data suggests that
- o Paragraph 2/Line 4/5: using a high content microscopy
- o Paragraph 2/Line 12: treatment respectively with Iovastatin and RO-3306, respectively (Fig.S6B).
- o Paragraph 3/Line 1: Altogether, this data indicates that, while...

- Page 10:

- o Paragraph 1/Line 1: ...SETX was found to be recruited to DSBs...
- o Figure 5B: is the increase of ssDNA in SETX depleted samples at 1626bp significant? Please comment. Also, please state statistics test used to determine significance in the Figure legend.
- o Paragraph 2/Line 3: You show NHEJ/ GC and SSA. What about MMEJ?
- o Figure 5C: Are the changes in SETX depleted samples significant? Again, please state statistics test carried out to determine the p-value.
- o Paragraph 3/Line 1: Collectively, this data indicates that although recruited to DSBs...

- Page 11:

- o Paragraph 1/Line 2: ... of closely clustered DBSs. found as able to cluster together
- o Paragraph 1/Line 5/6: This data indicates
- o Paragraph 2/Line 2: R-Loops removal
- o Paragraph 2/Line 2: SETX, a
- o Paragraph 2/Line 3: RNA:DNA hybrids helicase
- o Paragraph 2/Line 3/4: ...accumulate on chromatin flanking DSBs, exclusively at actively transcribed loci.
- o Paragraph 2/Line 5: ...also found to be recruited to DSBs....
- o Paragraph 2/Line 6: SETX might further
- o Paragraph 3/Title : R-Loops accumulation
- o Paragraph 3/Line 3: R-Loops accumulation
- o Paragraph 3/Line 4: what do you mean with "their" general decrease?

- Page 12:

- o Paragraph 1/Line 2: ... as well as in chromatin proximally flanking DSBs...
- o Paragraph 1/Line 2/3: .., "although being not affected at distance of the break, in the entire gH2AX domain" this sentence is unclear. Please rephrase.
- o Paragraph 1/Line 8: slowly elongating RNAPolIII, and
- o Paragraph 1/Line 9: "spurious" meaning unclear in this context. Maybe "illegitimate" or "false"
- o Paragraph 1/Line 12: R-Loops accumulation
- o Paragraph 1/Line 14: "pausing" maybe "stalling" a more appropriate word?

- Page 13:

- o Paragraph 1/Line 3: dsRNA was found
- o Paragraph 1/Line 3: R-Loops enriched
- o Paragraph 1: Last sentence unclear. Missing words? Or possible wrong structure. Please rephrase.
- o Paragraph 2/Line 1: Who/ what do you refer to as "they"?
- o Paragraph 2/Line 5: remodelling
- o Paragraph 2/Line 5: Notably, the TIP60 complex
- o Paragraph 2/Line 8: "timely repair" Data shows no change on repair kinetics.
- o Paragraph 3/Line 6: ... the damaged region to favor accessibility of repair proteins access

- Page 14:

- o Paragraph 2/Line 1: R-Loops removal
- o Paragraph 2/Line 2: recruit helicases including
- o Paragraph 2/Line 4: strongly supports this
- o Paragraph 2/Line 5: R-Loops removal
- o Paragraph 2/Line 11: chromatin fiber, which

- o Paragraph 2/Line 14: an increased mobility of DNA ends mobility triggered by the enhanced
- o Paragraph 3/Line 6: FANCM, which
- o Paragraph 3/Line 7: R-Loops removal

- Page 15:

- o Paragraph 1/Line 2: In this regard, it is

Figure 6 D:

R-loops accumulation around DSB

Figure S4:

Correct scientific notation is 50nM/70nM/100nM concentration for Etoposide, not 0.05 μ M

Figure5/6: What statistics test was used to determine p-values?

Reviewer #2:

Remarks to the Author:

The manuscript by Cohen et al. investigates the role of R-loops and the SETX helicase in DSB repair. The authors report that SETX and R-loops are enriched around DSBs within transcribed gene loci. Depletion of SETX leads to increased H2AX accumulation around DSBs, enhanced levels of illegitimate re-joining of distant DNA ends and to an overall impairment of survival following DSBs.

The topic of the manuscript is very interesting and timely. Recent studies have reported the presence of ncRNAs and/or DNA-RNA hybrids around DSBs in yeast and other organisms and have suggested various models for the formation and potential role of these RNA transcripts and RNA processing activities. Further, high quality studies are necessary to confirm and extend these findings and to mechanistically understand the role of RNA and RNA processing activities in the DNA repair process. However, in its current form, the manuscript raises a lot of questions and I am not convinced that the major conclusions are well supported by the experimental evidence.

Major concerns:

1. As the authors mention, the AsiSI-induced DSB system can access only about ~6% of the potential cleavage sites (80 out of 1211 sites), which seems to be a very special subset of genomic locations. As far as I know, most of the accessible sites (or probably all??) are located in active promoter regions. While I understand that this system was used previously in other studies to investigate DSB repair and it was partially characterized, I don't know if it is clearly understood why this subset of cleavage sites is accessible and why the rest of the sites are not accessible. This is particularly important for this study, since the authors claim that R-loops appear only in transcribed regions and some of their main conclusions are based on this observation.

- How many sites (out of the 80 that are cleaved) are actually non-transcribed (according to the Pol II ChIP-seq data)? If only a handful of sites are "non-transcribed", does this allow the authors to draw the strong conclusion they made about transcribed and non-transcribed regions? (see also 2nd major concern)

- What about the remaining 1130 sites? Are they all in intergenic regions and at silenced heterochromatic regions, or are they also in active promoter or gene regions? Figure 2B shows that the DRIP-seq read counts are significantly lower at the uncut sites than at cut sites (the difference between uncut and cut is more robust than the difference between cut -4OHT and cut+4OHT), suggesting that indeed, uncut sites are generally less transcribed (or non-transcribed, if this level of DRIP-seq reads represents the background signal) than cut sites.

While there is no perfect system for DSB induction, the limitations of each system have to be considered and clearly discussed.

2. Methods section states that "For RNA pol II ChIP, 25 μ g of chromatin was immunoprecipitated with 2 μ g of anti-RNA polymerase II CTD repeat YSPTSPS (phospho S2) (Abcam, ab5095)". I

double-checked to exclude that this was a typo, but indeed, ab5095 is a Pol II-Ser2-P antibody. This is my biggest concern with this manuscript and admittedly it shook my trust in the rest of the data. How do the authors explain the peaks around promoter regions observed with this antibody in the selected genes (Fig1E, Fig2A and in Supplementary figures), since, Pol II-Ser2 is normally not phosphorylated around promoter regions. Pol II-Ser-2 is phosphorylated in the gene body, especially at the 3' end of the genes.

- Did the authors classify if a gene was expressed or not based on the appearance of this uncharacteristic peak at the promoter region and did they base their further analyses on this classification? This is not compatible with anything I know about Pol II transcription and CTD phosphorylation, and it is in sharp contrast with any Pol II-Ser2-P dataset that I know of, including our own results. Theoretically the authors should have uploaded their datasets to GEO or other databank and provided accession numbers for the reviewers. I would need to see the original data (e.g. in bigwig format) to be able to assess the quality of the datasets and to consider whether the authors' interpretation is valid. This major concern leads me to question the entire manuscript.

- Is it possible that the Pol II-Ser2-P peak before the promoters represent an unusually strong "Promoter Upstream transcript (PROMPT)" transcription in this special subset of genomic loci? If yes, would this influence the interpretation of the data? PROMPTs are very short, unstable transcripts. E.g. Figure 2A shows the ASXL1 gene, with typical 5' and 3' DNA-RNA hybrid peaks (DRIP(S9.6)-4OHT), but there is no sign of transcription of this gene above background signal in the RNAPolIII-4OHT except the uncharacteristic Pol II-Ser2-P peak upstream of the TSS, which would most likely overlap with the PROMPTs of this promoter.

3. The Methods section mentions that the authors carried out a No Antibody control ChIP seq. This is a necessary control to exclude that peaks around DSBs represent some kind of artefact around DNA breaks which might influence the library preparation. However, this negative control is not mentioned elsewhere in the manuscript and data is not shown in Figures or Supplementary figures. It would be necessary to include this extra line of negative controls e.g. in Figure S1 B to convincingly rule out the above-mentioned possibility.

4. It is clear that DSBs can be repaired by two very distinct mechanisms: NHEJ, or HR-mediated repair. The authors are assaying R-loops and transcription around DSBs, but the potential role and abundance of these transcripts/hybrids might be very different between NHEJ and HR. NHEJ is known to inhibit transcription around DSBs by the DNA protein kinase (DNAPK), which is recruited to DSBs by the key NHEJ factor, Ku80-Ku70. As far as I know, there is no clear experimental evidence to support transcriptional inhibition during HR-mediated DSB repair. However, the authors try to make conclusions and models to describe a "general DSB-repair mechanism", which, in my opinion, doesn't exist. Are they referring to NHEJ or HR-repair? Since the experiments are done in unsynchronized cells, the experiments mostly assay NHEJ, but the authors don't discuss this and they also put their results in the wrong context, e.g. by comparing their results to studies in *S. pombe*, a model system which is using almost exclusively HR-mediated DSB repair.

OTHER COMMENTS:

5. Topoisomerase inhibition leads to R-loop accumulation and SETX2 is involved in unwinding R-loops, so the sensitivity of siSETX2 can also be explained by this.

6. Fig 3C – can the authors show that pre-treatment with DRB does not decrease the expression of AsiSI? Could they also include the siCTR+4OHT+DRB picture and quantification?

7. I am not sure how the authors interpret the increased γ H2AX signal in siSETX+4OHT cells. Is the increased signal the result of an increased number of foci or increased amount of γ H2AX at the foci? Number of foci should correlate with the number of DSBs and the length of time the foci are visible – both of which was shown not to change in siSETX. If the increased signal is indeed the result of an increased amount of γ H2AX at DSBs - what does that mean? Does this represent impaired repair or enhanced repair?

8. Fig5B - the sensitivity of this experiment is questionable, it seems that the values are very close to background (CTRL-4OHT and SETX-4OHT), especially at the more distant sites. This is not very surprising, because likely only a small percentage of cells are in G2/S using HR and/or the cleavage efficiency is low. Although the figure legend indicates ssDNA(%), the data seems to be normalized to CTRL-4OHT, so how does this represent % of ssDNA? The authors conclude that the changes in ssDNA levels are not significant in the siSETX cells, so SETX is not involved in strand resection. However, if the background level is 1 and the measured values are around 2, a 2x increase will hardly be ever significant, although a 2x increase in ssDNA length would be biologically highly significant. See, for example, DSB-KDELR3 at 1626 bp and the similar trend in DSB-ASXL1. I don't think that the authors can conclude anything from these experiments.

Response to referees

First we would like to thank the referees for their thorough work on our manuscript and before our point by point response we would like to briefly mention what has been included in the revised manuscript:

- We now show that senataxin depletion increases R-loops accumulation at one DSB (by DRIP-qPCR, Fig. 2E), directly implicating senataxin in the processing of DSB-induced R-loops.
- We now show that senataxin depletion decreases Rad51 foci formation and increases 53BP1 (Figure 5). Increase in 53BP1 was partially rescued by a prior treatment with transcription inhibitors DRB or cordycepin (Figure S5)
- We report the translocation analysis on 2 more translocation events (for a total of 4 events analyzed), which are increased upon senataxin depletion
- We now show that RNaseH1 overexpression partially rescues the enhanced translocation observed in senataxin depleted cells, highlighting a direct role of R-loops in this process.
- We performed again the RNA PolII S2P ChIP-seq profile using a more specific antibody (see Figure for referee 2) and backed up these data by performing total RNA PolII ChIP-seq as well as RNA-seq in DivA cells (prior DSB induction).
- These new datasets confirmed nicely the strong accumulation of RNA:DNA hybrids and senataxin at damaged active genes.
- However, using RNA-seq (a highly sensitive method to measure transcriptional activity) we were also able to observe a handful of sites with no signs of apparent transcription where low level of R-loops accumulate following DSB induction but without senataxin recruitment. We now show some examples of these untranscribed loci in Figure S2C-D, along with an example of untranscribed locus that does not accumulate R-loop following breakage (Figure S2E) which shows that this does not represent a general phenomenon.
- We have therefore changed the discussion to mention that both RNA polII stalling and at least in few instances, *de novo* transcription may contribute to R-loops formation.

Reviewer #1 (Remarks to the Author):

Summary:

Using their previously developed cell line in which up to 80 sequence-specific DSBs

cells can be induced at specific sites, mainly active transcription units, in the genome, Cohen et al, assessed the role of R-loops in DSB response. Using this system, the authors showed that R-loops and the likely RNA:DNA helicase Senataxin (SETX) accumulate on chromatin flanking DSBs induced within transcribed genes. Interestingly, SETX depletion did not affect resection kinetics at DSBs, had minimal affect on DSB repair pathway choice and did not delay the kinetics of DSB repair. Surprisingly, and through a yet to characterised mechanism, they observed increased joining of distant DSBs to form translocations. Therefore, this study elucidates a new role for Senataxin and R-loops in regulating the fidelity of in DSB repair in mammalian cells. It also provides a possible resolution to the controversy about whether small dsRNAs are produced at DSBs. As small amounts of dsRNA has previously been reported to occur at transcription termination sites, which are also enriched in R-loops, the authors suggest that the so-called DDRNAs previously reported by others could simply result from pausing of RNA polIII at DSBs. In this respect pausing at DSBs really corresponds to premature transcription termination at DSBs within active transcription units.

First we thank the reviewer for his/her thorough work on our manuscript and his/her positive comments.

Major points:

1. To assess the role of SETX in the choice or repair pathway (NHEJ versus HR), they authors use reporter systems based on GFP expression following DSB repair (Figure 5). However, it is not clear if R-loops are formed in these systems, as not all transcription units are known to produce R-loops. The authors should discuss whether formation of R-loops was previously shown in these specific reporter systems and, if not, either confirm R-loop formation experimentally or discuss this caveat to their experiments. If R-loops are not formed, then using these repair systems would not address the role of SETX in the choice of repair pathways. This could in fact understate the role of SETX in DSB repair pathway choice.

We agree that the reporter systems used may not be suitable to study the function of SETX in repair pathway choice, due to the unknown status of R-Loops at the reporter loci, and we thank the referee for pointing this out. Hence to strengthen these data we further performed immunostaining of Rad51 in control and senataxin depleted cells. We could observe a decrease in rad51 foci formation, together with an increased 53BP1 signal. These data are now presented Fig. 5A/B and the assays with the reporter constructs were moved to Fig. 5C.

2. The authors have established an elegant assay to examine translocation events that occur between DSBs whose precise position on chromosomes is known (Figure 6). This provides sequence resolution of translocation events. It is an original approach presented for the first time in this paper. The data are strong but it would be useful to validate this method using an alternative technique, such as FISH. If the translocation events measured occur at a reasonable frequency, at least a few percent of the population, it might be possible to detect using FISH. Alternatively, if these events are rare it might be too much to expect their detection by this technology.

We thank the reviewer for his/her positive comment on the translocation assay. We agree that it would be interesting to validate it using another method, especially to have a clear estimation of the occurrence of such events. We are actively pursuing this objective. Our preliminary results using digital droplet PCR (ddPCR) indicate that in control cells, each translocation occurs at best in only 1/300 cells. Notably this measured frequency is in agreement with translocation frequency measured between I-SceI induced DSBs (Roukos et al, 2013). Given this very low frequency, FISH analysis seems compromised (unless using high throughput microscopy which we are considering). We are still optimizing the conditions for ddPCR and working on other means to detect these translocations, but given the very strong time constraints on this manuscript, we do not think we could include these data in a timely manner. However, we have now included other translocations in Figure S6A.

3. Should include a SETX cDNA rescue of the SETX knockdown as a control for one of the key experiments experiments of the paper, e.g. the translocation assay.

We agree with this reviewer, that ideally this experiment should have been performed. Once more, given the time constraint we could not perform it in due time. However, we now show that the overexpression of RNaseH in senataxin depleted cells partially rescues the increase of translocation (Figure S6C). Of note, most of the experiments (including the translocation assay) were done using both a single SETX siRNA (SETX#2) but also a pool of siRNA (SETX Smart Pool, see Figure S6B), which we think further strengthen the data.

4. Overall the English needs to be improved by careful editing (see some suggestions below).

Minor points:

1. Legend to Figure 1. Regarding the BLESS dataset, it is not appropriate to refer to another manuscript in revision with a legend.

We apologize and removed the reference from the figure Legend.

2. Spell out "DSB Inducible via AsiSI" before first use of jargon acronym, DIVA.

This has been changed accordingly

3. Figure 1A. Provide zoomed-in view of both genome browser shots around DSBs. Histogramme for RBMXL1 gene looks minor in this view.

We followed the advice of this referee and now also show zoomed in screenshots

4. Figure 1E could usefully include a positive control, something recruited to both DSBs. However, if such a data set is unavailable then this is not a major flaw overall.

We now show the XRCC4 binding at both sites presented Figure 1E in Figure S1C

5. Page 7, first line. Important to state fold enrichment here. It appears to be statistically significantly but what is the fold enrichment measured. Appears to be quite small, 20% increase over uncut at best. Overall the data in Figure 2 is very nice.

Fold enrichment has been mentioned in the text

6. Figure 3B. Show quantification of gH2AX signal from -4OHT cells.

This has been included

7. Page 9, 4 lines from top. More accurate to state that DRB "partially rescued the increased gH2AX foci observed following SETX depletion."

This has been modified accordingly

8. Page 10, 10 lines down. "associated with a similarly mild increase of NHEJ" .

This has been modified accordingly

9. Regarding the required thorough editing of the language of the manuscript. One of the researchers in my lab that I consulted for advice on this manuscript went to the trouble of listing some suggested changes which I reproduce here for the benefit of the authors .

We thank very much this referee and his/her researcher, for the extensive proofreading and suggested changes. Indeed, it is very helpful to us. We have generally followed the proposed edits.

Reviewer #2 (Remarks to the Author):

The manuscript by Cohen et al. investigates the role of R-loops and the SETX helicase in DSB repair. The authors report that SETX and R-loops are enriched around DSBs within transcribed gene loci. Depletion of SETX leads to increased H2AX accumulation around DSBs, enhanced levels of illegitimate re-joining of distant DNA ends and to an overall impairment of survival following DSBs.

The topic of the manuscript is very interesting and timely. Recent studies have reported the presence of ncRNAs and/or DNA-RNA hybrids around DSBs in yeast and other organisms and have suggested various models for the formation and potential role of these RNA transcripts and RNA processing activities. Further, high quality studies are necessary to confirm and extend these findings and to mechanistically understand the role of RNA and RNA processing activities in the DNA repair process. However, in its current form, the manuscript raises a lot of questions and I am not convinced that the major conclusions are well supported by the experimental evidence.

First we thank the reviewer for his/her thorough work on our manuscript, and his/her interest in our study.

Major concerns:

1. As the authors mention, the AsiSI-induced DSB system can access only about ~6% of the potential cleavage sites (80 out of 1211 sites), which seems to be a very special subset of genomic locations. As far as I know, most of the accessible sites (or probably all??) are located in active promoter regions.

While I understand that this system was used previously in other studies to investigate DSB repair and it was partially characterized, I don't know if it is clearly understood why this subset of cleavage sites is accessible and why the rest of the sites are not accessible. This is particularly important for this study, since the authors

claim that R-loops appear only in transcribed regions and some of their main conclusions are based on this observation.

- How many sites (out of the 80 that are cleaved) are actually non-transcribed (according to the Pol II ChIP-seq data)? If only a handful of sites are non-transcribed, does this allow the authors to draw the strong conclusion they made about transcribed and non-transcribed regions? (see also 2nd major concern)

- What about the remaining 1130 sites? Are they all in intergenic regions and at silenced heterochromatic regions, or are they also in active promoter or gene regions? Figure 2B shows that the DRIP-seq read counts are significantly lower at the uncut sites than at cut sites (the difference between uncut and cut is more robust than the difference between cut -4OHT and cut+4OHT), suggesting that indeed, uncut sites are generally less transcribed (or non-transcribed, if this level of DRIP-seq reads represents the background signal) than cut sites. While there is no perfect system for DSB induction, the limitations of each system have to be considered and clearly discussed.

Indeed AsiSI is not able to cleave methylated sequences (Iacovoni et al, 2010; Aymard et al, 2014). Hence a fraction of AsiSI sites are refractory to cleavage due to their methylation state (Figure for referee 1A). Given that some uncleaved AsiSI sites do not exhibit methylation we believe that additional features also decrease AsiSI efficiency such as chromatin compaction, nucleosome positioning or potential SNP that could disrupt the AsiSI sites. Nevertheless, we can still analyze repair events on no less than 80 DSBs which provide a statistical power that is yet unachieved with other systems.

Very importantly, even if we agree that there is a bias toward actively transcribed loci, the AsiSI cut sites population is not restricted to active promoters. We provide on Figure for referee 1B, the distribution of cut AsiSI sites based on the ChromHMM classification of genomic loci (ENCODE). This shows that some cut AsiSI sites are not in the "active promoter" class. Moreover, we show Figure for referee 1C, the distribution of the local RNA PolIII S2P level for all AsiSI sites. Cut AsiSI sites are depicted in red, further showing that all DSBs are not located within transcribed loci. Finally we show multiple examples in this manuscript of robustly induced DSBs in loci that do not exhibit neither RNA PolIII S2P nor RNA-seq signal, indicating that they lie within transcriptionally silent loci.

Altogether, these data clearly demonstrate that we are able to compare damage response both at actively transcribed and silent regions

We agree that among the 80 DSBs, untranscribed cleaved loci are less frequent, but apart from that, there are no good reason suggesting that we cannot draw conclusion

from these sites. Of note most of the sequence-specific DSB studies in the field are performed by inducing either very few DSBs (I-PpoI) or a single DSB (on a transgenic construct (I-SceI) or at one endogenous location (CRISPR/cas9 induced DSB, or Fok1 induced DSB)). Nevertheless, conclusions could be drawn from such studies (see for example Shanbag et al, Cell 2010, Tang et al, NSMB 2013, Pankotai, NSMB 2010, Ohle et al, 2016 í). While we agree that the low amount of DSB induced in silent loci in our cells may limit the statistical power obtained at such loci, we still believe we can interpret these data. More specifically senataxin recruitment was never observed at transcriptionally silent sites. For R-loops, both cases (low R-loops formation or not) were observed at transcriptionally silent sites, hence further studies should be performed using other systems to thoroughly analyze their formation at silent loci. This is now mentioned in the discussion.

2. Methods section states that ðFor RNA pol II ChIP, 25 g of chromatin was immunoprecipitated with 2 g of anti-RNA polymerase II CTD repeat YSPTSPS (phospho S2) (Abcam, ab5095)ö. I double-checked to exclude that this was a typo, but indeed, ab5095 is a Pol II-Ser2-P antibody. This is my biggest concern with this manuscript and admittedly it shook my trust in the rest of the data. How do the authors explain the peaks around promoter regions observed with this antibody in the selected genes (Fig1E, Fig2A and in Supplementary figures), since, Pol II-Ser2 is normally not phosphorylated around promoter regions. Pol II-Ser-2 is phosphorylated in the gene body, especially at the 3ø end of the genes.

- Did the authors classify if a gene was expressed or not based on the appearance of this uncharacteristic peak at the promoter region and did they base their further analyses on this classification? This is not compatible with anything I know about Pol II transcription and CTD phosphorylation, and it is in sharp contrast with any Pol II-Ser2-P dataset that I know of, including our own results. Theoretically the authors should have uploaded their datasets to GEO or other databank and provided accession numbers for the reviewers. I would need to see the original data (e.g. in bigwig format) to be able to assess the quality of the datasets and to consider whether the authorsøinterpretation is valid. This major concern leads me to question the entire manuscript.

- Is it possible that the Pol II-Ser2-P peak before the promoters represent an unusually strong ðPromoter Upstream transcript (PROMPT)ö transcription in this special subset of genomic loci? If yes, would this influence the interpretation of the data? PROMPTs are very short, unstable transcripts. E.g. Figure 2A shows the ASXL1 gene, with typical 5øand 3øDNA-RNA hybrid peaks (DRIP(S9.6)-4OHT), but there is no sign of transcription of this gene above background signal in the RNAPolIII-

4OHT except the uncharacteristic Pol II-Ser2-P peak upstream of the TSS, which would most likely overlap with the PROMPTs of this promoter.

We agree with this referee that in this set of data, in addition to the classical 3' end accumulation, a peak was also found at promoters. This is due to the antibody commercialized by Abcam that is not fully specific and also recognizes other forms of RNA PolIII (see Figure for Referee 2). Of note, ChIP-Seq data published earlier this year by the lab of Dr F. Dødda di Fagagna and using the exact same antibody (from Abcam) and cell line (DIvA) showed a very similar pattern (Iannelli et al, 2017) (see Figure for referee 2A/B).

Hence, we have repeated the ChIP-seq experiment using the antibody directed against RNA PolIII S2P from Chromotek, which provided much better results (see Figure for referee 2A/B). We have also backed up this analysis by performing Total RNA polIII ChIP-seq (independent of any phosphorylation) and RNA-seq in untreated DIvA cells. These three new datasets are now included in the revised manuscript. Similar data about senataxin recruitment were obtained when determining the transcriptional status of a given gene, using RNA-seq, total RNA PolIII ChIP-seq or RNA PolIII S2P ChIP-seq (Fig. 1D, Fig. S1A). However, using the RNA-seq dataset which provides a very sensitive way to measure transcriptional activity, we could observe very low levels of R-loops formation at some sites totally devoid of transcriptional activity (Fig. S2C-D), and we have now modified the manuscript accordingly. Hence, we thank this referee for the opportunity to include better data in our manuscript, leading to more accurate interpretations.

The raw data are currently under submission to ArrayExpress, but the referee can download our bigwig. files at the following http address

<https://mycore.core-cloud.net/index.php/s/X9rkUWNesu8q4UZ>

Pwd : Zrf31000-GLTSP

3. The Methods section mentions that the authors carried out a No Antibody control ChIP seq. This is a necessary control to exclude that peaks around DSBs represent some kind of artefact around DNA breaks which might influence the library preparation. However, this negative control is not mentioned elsewhere in the manuscript and data is not shown in Figures or Supplementary figures. It would be necessary to include this extra line of negative controls e.g. in Figure S1 B to convincingly rule out the above-mentioned possibility.

We apologize for the misstatement. While we always perform no antibody controls in ChIP experiments to assess the quality of the ChIP, it was not sent for sequencing.

However, we provide now the profile for histone H3 (see examples Fig. S1C) which clearly shows that senataxin peaks at DSBs while H3 does not (showing that senataxin peak does not arise from an artefact in library preparation). Additionally, senataxin is not found at other DSBs which yet display strong cutting efficiency (Fig. 1E, Fig. S2), further strengthening this point.

4. It is clear that DSBs can be repaired by two very distinct mechanisms: NHEJ, or HR-mediated repair. The authors are assaying R-loops and transcription around DSBs, but the potential role and abundance of these transcripts/hybrids might be very different between NHEJ and HR. NHEJ is known to inhibit transcription around DSBs by the DNA protein kinase (DNAPK), which is recruited to DSBs by the key NHEJ factor, Ku80-Ku70. As far as I know, there is no clear experimental evidence to support transcriptional inhibition during HR-mediated DSB repair. However, the authors try to make conclusions and models to describe a "general DSB-repair mechanism", which, in my opinion, doesn't exist. Are they referring to NHEJ or HR-repair? Since the experiments are done in unsynchronized cells, the experiments mostly assay NHEJ, but the authors don't discuss this and they also put their results in the wrong context, e.g. by comparing their results to studies in *S. pombe*, a model system which is using almost exclusively HR-mediated DSB repair.

We apologize but we found this comment confusing and we are not sure how to interpret the question. We previously reported that actively transcribed genes tend to be repaired by HR (in G2) while untranscribed loci do not rely on Rad51 for repair. In this manuscript we also show that senataxin is more recruited at DSB induced in active genes. We now also show as New Fig. S1D that senataxin is indeed also more recruited at sites bound by Rad51 than sites unbound by Rad51. Hence our data point toward a model where both Rad51 and senataxin exhibit a preference for repairing active genes.

Of note transcriptional repression of broken genes has also been previously linked to the proper completion of HR repair. For example, the chromatin reader ZMYND8 which is recruited at DSB to sustain NURD recruitment and to promote transcription repression, also promotes HR (Gong et al, 2015). Similarly KDM5A sustains both HR and transcriptional repression (Gong et al, 2017). Moreover we now provide additional data showing that senataxin depletion also impedes Rad51 foci formation further strengthening the finding that senataxin rather contributes to HR at active genes.

OTHER COMMENTS:

5. Topoisomerase inhibition leads to R-loop accumulation and SETX2 is involved in

unwinding R-loops, so the sensitivity of siSETX2 can also be explained by this. We agree with this comment, and cannot exclude this hypothesis. However, given that etoposide mostly induces DSB in active promoters, together with the sensitivity of D1vA cells and not upon IR, we think this results equally strengthens the idea that senataxin promotes cell survival upon active gene breakage.

6. Fig 3C ó can the authors show that pre-treatment with DRB does not decrease the expression of AsiSI? Could they also include the siCTR+4OHT+DRB picture and quantification?

We now show multiple data showing that 53BP1 and gammaH2AX foci are normally induced upon DRB or cordycepin pretreatment, strongly suggesting that available AsiSI is sufficient to properly induce DSB upon transcriptional inhibition

7. I am not sure how the authors interpret the increased H2AX signal in siSETX+4OHT cells. Is the increased signal the result of an increased number of foci or increased amount of H2AX at the foci? Number of foci should correlate with the number of DSBs and the length of time the foci are visible ó both of which was shown not to change in siSETX. If the increased signal is indeed the result of an increased amount of H2AX at DSBs - what does that mean? Does this represent impaired repair or enhanced repair?

We do not think that increased H2AX signal in SETX-depleted cells corresponds to an increase in foci number, since as pointed out by the referee, repair kinetic is not affected in senataxin-depleted cells. Hence we think that γ H2AX increases within the foci. As now mentioned in the discussion (p15), one possibility may be that R-loops removal and/or senataxin promote ATM activity and enhance γ H2AX domain formation. Alternatively senataxin depletion could also modify chromatin structure in *cis* to the DSB, rendering it more permissive to γ H2AX establishment. Many hypotheses can be considered at this stage but this falls beyond the scope of this study.

8. Fig5B - the sensitivity of this experiment is questionable, it seems that the values are very close to background (CTRL-4OHT and SETX-4OHT), especially at the more distant sites. This is not very surprising, because likely only a small percentage of cells are in G2/S using HR and/or the cleavage efficiency is low. Although the figure legend indicates ssDNA(%), the data seems to be normalized to CTRL-4OHT, so how does this represent % of ssDNA? The authors conclude that the changes in ssDNA levels are not significant in the siSETX cells, so SETX is not involved in strand resection. However, if the background level is 1 and the measured values are around 2, a 2x increase will hardly be ever significant, although a 2x increase in

ssDNA length would be biologically highly significant. See, for example, DSB-KDELR3 at 1626 bp and the similar trend in DSB-ASXL1. I don't think that the authors can conclude anything from these experiments.

We clearly show here that senataxin is not required for resection. Using siRNA against CtIP and against SETD2 we were able to detect very strong defect in resection using the exact same assay (Pfister et al, Cell Reports, 2014), so we believe this assay is sensitive enough to detect strong changes in resection. However, we agree that this resection assay only monitors changes up to 1.6kb and hence, we cannot exclude a possible effect of senataxin in regulating long range resection. This is now mentioned in the discussion.

The ssDNA was normalized to the signal without OHT in control cells which allows us to compile 4 independent experiments. We apologize for the wrong annotation in the previous manuscript, and this has been modified on the y axis in Figure 5. The referee can find below a typical experiment (DSB-KDELR3) which is not normalized.

Figure1:
 A. Dotplot representing MethylCap-seq read count obtained in U2OS (Deplus et al, Cell Reports 2014), on a 200bp window for each the 1211 predicted AsiSI sites in the human genome. Sites are sorted by decreasing signal. Cut sites are indicated in red.
 B. The 80 DSBs were compared with the chromatin state segmentation track from hESC and K562 cells (Broad ChromHMM, <http://rohsdb.cmb.usc.edu/GBshape/cgi-bin/hgTrackUI?db=hg19&g=wgEncodeBroadHmm>). The proportion of DSBs lying within active promoters (red) or other loci (grey) are shown.
 C. Dotplot representing RNA PolIII-S2P read count in a 10 kb window for the 1211 predicted AsiSI sites in the human genome. Sites are sorted by decreasing signal. Cut sites are indicated in red.

A

B

Figure 2:
 A. Genome browser screenshots showing the RNA-PolIII-S2P profile obtained with the chromotek antibody (our revised manuscript), with the Abcam antibody reference 5095 (our original submission) and reported earlier this year in the same cell line (DIvA) by the laboratory of Dr d'adda Di Fagagna (Iannelli et al, Nat Comm 2017) using the same Abcam 5095 antibody.

B. Averged profiles of RNA PolII-S2P on all genes on the genome, for each datasets.

Although showing enrichment at 3' end as expected, the abcam antibody displays some unspecificity, likely recognizing other forms of RNA PolIII (either unmodified or phosphorylated on other residues). The chromotek antibody in contrast, shows a very specific signal at TTS.

Reviewers' Comments:

Reviewer #1:

Remarks to the Author:

I have now reviewed of the revised manuscript of Cohen et al and I am satisfied that it is highly suitable for publication in Nat Comms. This work is important. It demonstrates a role for Senataxin in counteracting the illegitimate rejoining of DSBs induced in actively transcribed regions of the genome. Their data suggest that senataxin is required to removed R-loops formed directly at the break. This Transcription Coupled DSB repair is no doubt particularly critical for repair of transcriptionally highly active neurons explaining the phenotype of patients with defective senataxin.

The manuscript, both results and discussion sections, has been substantially improved by peer review.

In their rebuttal document the authors highlight the inclusion of substantial new data. They also highlight the advantages (statistical power) of the AsiSI system, while accepting that there is no perfect system. I agree with their justification of why the addition of further work would be beyond the scope of the present study.

Minor point. Page 6, line 7. The word "devoid" should be changed. Perhaps the authors would prefer to use 'deserts'?

Figure 2A. Both examples are with DSB induced at or very close to TSS. Do the authors have a suitable example with a DSB within a gene body?

Reviewer #2:

Remarks to the Author:

The authors have answered all my questions and concerns. The manuscript and the presented data are significantly improved and I recommend it for publication.

REVIEWERS' COMMENTS

Reviewer #1 (Remarks to the Author):

I have now reviewed of the revised manuscript of Cohen et al and I am satisfied that it is highly suitable for publication in Nat Comms. This work is important. It demonstrates a role for Senataxin in counteracting the illegitimate rejoining of DSBs induced in actively transcribed regions of the genome. Their data suggest that senataxin is required to removed R-loops formed directly at the break. This Transcription Coupled DSB repair is no doubt particularly critical for repair of transcriptionally highly active neurons explaining the phenotype of patients with defective senataxin.

The manuscript, both results and discussion sections, has been substantially improved by peer review.

In their rebuttal document the authors highlight the inclusion of substantial new data. They also highlight the advantages (statistical power) of the AsiSI system, while accepting that there is no perfect system. I agree with their justification of why the addition of further work would be beyond the scope of the present study.

We thank the referee for his work on our manuscript

Minor point. Page 6, line 7. The word "devoid" should be changed. Perhaps the authors would prefer to use 'deserts'?

This sentence has been rephrased

Figure 2A. Both examples are with DSB induced at or very close to TSS. Do the authors have a suitable example with a DSB within a gene body?

Such an example is now shown as Figure S2C (DSB in the first intron of a gene)